# PLM-interact: extending protein language models to predict protein-protein interactions

Dan Liu [1], Francesca Young[1], Kieran D. Lamb [1], Adalberto Claudio Quiros[2,3], Alexandrina Pancheva[4], Crispin J. Miller [2,4], Craig Macdonald [3,5] ✉, David L. Robertson [1,5] ✉ & Ke Yuan [2,3,4,5] ✉

Computational prediction of protein structure from amino acid sequence alone has been achieved with unprecedented accuracy, yet the prediction of protein-protein interactions remains a challenge. Here, we assess the ability of protein language models (PLMs), routinely applied to protein folding, to be retrained for protein-protein interaction prediction. Existing models that exploit PLMs use a pre-trained PLM feature set, ignoring that the proteins are physically interacting. We propose PLM-interact, which goes beyond single proteins by jointly encoding protein pairs to learn their relationships, analogous to the next-sentence prediction task from natural language processing. This approach achieves state-of-the-art performance in a widely adopted cross-species protein-protein interaction prediction benchmark: trained on human data and tested on mouse, fly, worm, *E. coli* and yeast. In addition, we develop a fine-tuning method for PLM-interact to detect mutation effects on interactions. Finally, we report that the model outperforms existing approaches in predicting virus-host interaction at the protein level. Our work demonstrates that large language models can be extended to learn the intricate relationships among biomolecules from their sequences alone.

Proteins are the main structural components of cells and mediate biological processes by interacting with other proteins[1]. Disruption of these protein-protein interactions (PPIs), e.g., mediated by mutations, can underlie human disease[2]. In virology, PPIs are particularly important as viruses depend entirely on the host cell for replication, achieved mainly through specific interactions with host proteins. In response to infection, our immune system counteracts pathogens via targeted PPIs. Understanding PPI mechanisms offers the potential for developing novel therapy strategies for both human disease and pathogen infections[3]. Unfortunately, experimentally identifying PPIs is both costly and time-consuming, such that interaction datasets remain sparse with only a few species having comprehensive coverage[4,5].

Computational algorithms offer an efficient alternative to the prediction of PPIs at scale. Existing prediction approaches mainly leverage protein properties such as protein structures, sequence composition and evolutionary information[6–9]. Applying these features to pairs of proteins, classifiers have been trained using classical machine learning[10] and deep learning approaches[11]. Recently, protein language models (PLMs) trained on large public protein sequence databases have been used for encoding sequence composition, evolutionary and structural features[12–14], becoming the method of choice for representing proteins in state-of-the-art PPI predictors. A typical PPI prediction architecture uses a pre-trained PLM to represent each protein in a pair separately, then a classification head is trained for a

¹MRC-University of Glasgow Centre for Virus Research, Glasgow, United Kingdom. ²School of Cancer Sciences, University of Glasgow, Glasgow, United Kingdom. ³School of Computing Science, University of Glasgow, Glasgow, United Kingdom. ⁴Cancer Research UK Scotland Institute, Glasgow, United Kingdom. ⁵These authors jointly supervised this work: Craig Macdonald, David L Robertson, Ke Yuan. ✉e-mail: craig.macdonald@glasgow.ac.uk; david.l.robertson@glasgow.ac.uk; ke.yuan@glasgow.ac.uk

binary task that discriminates interacting pairs from non-interacting pairs[13,15] (Fig. 1a). Despite the use of PLMs in PPI prediction, identifying PPIs remains challenging.

The main issue is PLMs are primarily trained using single protein sequences, i.e., while they learn to identify contact points within a single protein[16], they are not 'aware' of interaction partners. In a conventional PLM-based PPI predictor architecture, a classification head is used to extrapolate the signals of inter-protein interactions by grouping common patterns of intra-protein contacts in interacting and non-interacting pairs, respectively (Fig. 1a). However, this strategy relies on the classification head being generalisable. Unfortunately, with the use of 'frozen' embeddings and a feedforward neural network being the dominant option, these classifiers have limited parameters to deal with complex interaction patterns.

To address the lack of inter-protein context in model pre-training, we propose PLM-interact, which directly models PPIs by extending and fine-tuning a pre-trained PLM, ESM-2[17]. PLM-interact (trained on human PPI data) achieves a significant improvement compared to other predictors when applied to mouse, fly, worm, yeast and *E. coli* datasets, and can be applied to virus-host PPI prediction. We also demonstrate that a fine-tuned version of PLM-interact can predict mutation effects on interactions.

## Results
### PLM-interact
To directly model PPIs, two extensions to the widely used PLM ESM-2[17] are introduced (Fig. 1b): (1) longer permissible sequence lengths in paired masked-language training to accommodate amino acid residues from both proteins; (2) implementation of "next sentence" prediction[18] to fine-tune all layers of ESM-2 where the model is trained with a binary label indicating whether the protein pair is interacting or not (see Methods for details). Our training task is, thus, a mixture of the next sentence prediction and mask language modelling tasks. This architecture enables amino acids in one protein sequence to be associated with specific amino acids from

another protein sequence through the transformer's attention mechanism.

The training of PLM-interact begins with the pre-trained large language model ESM-2. We fine-tune it for PPIs by showing it pairs of known interacting and non-interacting proteins. In contrast to similar training strategies in machine learning[18], we find that the next sentence prediction and mask language modelling objectives need to be balanced. We therefore conducted comprehensive benchmarking for different weighting options, before selecting a 1:10 ratio between classification loss and mask loss, combined with initialisation using the ESM-2 (with 650 M parameters), as this achieved the best performance (see Methods for details, Supplementary Fig. 1 and Supplementary Fig. 2).

### PLM-interact improves prediction performance
To examine the performance of PLM-interact, we benchmark the model against six PPI prediction approaches: TUnA[19], TT3D[13], Topsy-Turvy[20], D-SCRIPT[15], PIPR[6] and DeepPPI[21]. We use a multi-species dataset created by Sledzieski et al.[15]. Each model is trained on human protein interaction data and tested on five other species. The human training dataset in this multi-species dataset includes 421,792 protein pairs (38,344 positive interaction pairs and 383,448 negative pairs), human validation includes 52,725 protein pairs (4794 positive interaction pairs and 47,931 negative pairs) and the mouse, worm, fly and yeast test datasets each includes 55,000 pairs (5000 positives interaction pairs and 50,000 negative pairs), except for the *E. coli* test dataset, which includes 22,000 pairs (2000 positive interaction pairs and 20,000 negative pairs). The positive PPIs in these datasets are experimentally-derived physical interactions, while the negative pairs are randomly paired proteins not reported to interact.

PLM-interact achieves the highest AUPR (area under the precision-recall curve)[22] followed by TUnA[19] and TT3D[13] (Fig. 2b). Testing on mouse, fly and worm test species datasets, PLM-interact demonstrates AUPR improvements of 2%, 8% and 6% compared to TUnA[19], and 16%, 21% and 20% compared to TT3D[13], respectively. The predictions for

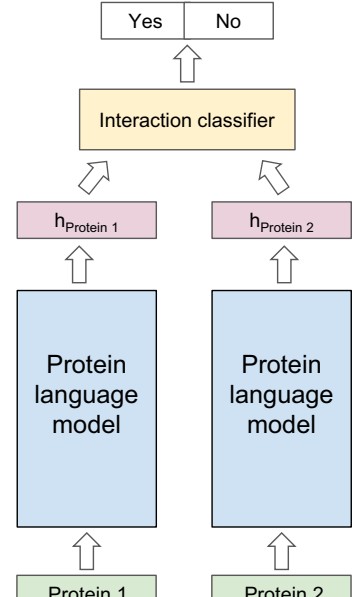

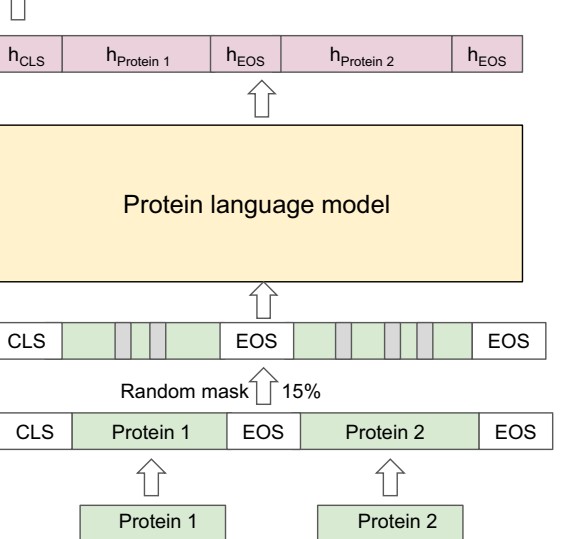

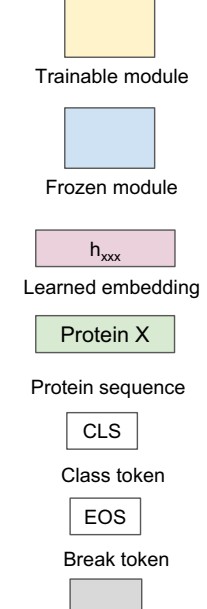

**a** Typical PPI prediction framework

**b** PLM-interact prediction framework

**Fig. 1 | A comparison of PLM-interact to an existing protein-protein interaction (PPI) prediction architecture. a** Typical PPI prediction models use pre-trained 'frozen' protein language models to extract single-protein embeddings with a trainable interaction classifier. **b** PLM-interact uses a protein language model with a longer context to handle a pair of protein sequences directly. Both the mask language modelling task and a binary classification task predicting interaction status are used to train the model.

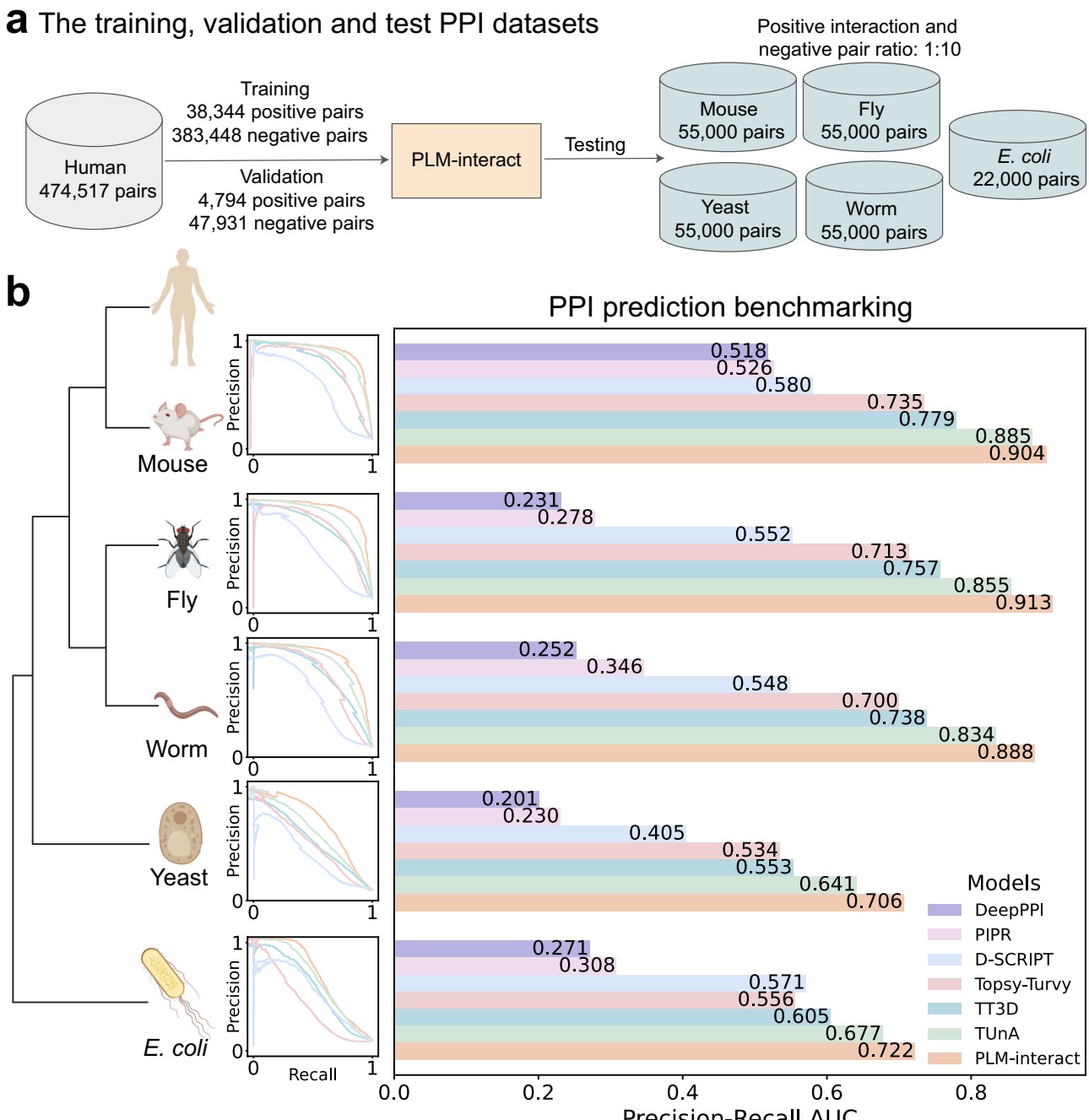

**Fig. 2 | The benchmarking results of PLM-interact compared with state-of-the-art protein-protein interaction (PPI) prediction models: PLM-interact achieves the highest PPI prediction performance. a** The data size of training, validation and test protein pairs. **b** The taxonomic tree of the training and test species is aligned with the precision-recall curve of each model on each test species. A bar plot of AUPR values illustrates the PPI prediction benchmark. The distribution of predicted interaction probabilities of positive and negative protein pairs for each PPI model is shown in Supplementary Fig. 3. All species icons in panel (**b**) are created in BioRender (Liu, D., 2025; https://BioRender.com/1ezsj7q). Source data are provided as a Source Data file.

yeast and *E. coli* PPIs are more challenging because they are more evolutionarily divergent from the human proteins used for training than the other species (see Fig. 2b). Our model achieved an AUPR of 0.706 on yeast, a 10% improvement over TUnA's AUPR of 0.641, and a 7% improvement on *E. coli* with an AUPR of 0.722; it also shows a 28% improvement over TT3D's AUPR of 0.553 on yeast and a 19% improvement over TT3D's AUPR of 0.605 on *E. coli*.

Importantly, the improvement in PLM-interact is due to its ability to correctly identify positive PPIs: Comparing the predicted interaction probabilities, PLM-interact consistently assigned higher probabilities of interaction to true positive PPIs. In contrast, other methods give lower interaction probability estimates in all held-out species. The distribution of predicted interaction probabilities of positive and negative protein pairs for each model is shown in Supplementary Fig. 3.

Next, we showcase five positive PPI instances, one for each test species, for which our model produces a correct prediction, and both TUnA and TT3D produce incorrect predictions (Fig. 3). These PPIs are necessary for essential biology processes including inducing leukaemia cell differentiation[23], dynein light chain roadblock[24], RNA

## Mouse
### Induced myeloid leukemia cell differentiation protein Mcl-1 homolog and monooxygenase COQ6 and Translationally-controlled tumor protein

## Fly
### Dynein light chain roadblock and AT23443p

## Worm
### Mediator of RNA polymerase II transcription subunit 15 and 19

**Mouse**

TT3D: 0.204    PLM-interact: 0.728
TUnA: 0.083

0        0.5         1
Predicted interaction probability

**Fly**

TT3D: 0.078    PLM-interact: 0.915
TUnA: 0.425

0        0.5         1
Predicted interaction probability

**Worm**

TT3D: 0.008    PLM-interact: 0.792
TUnA: 0.054

0        0.5         1
Predicted interaction probability

## Yeast
### Mitochondrial import receptor subunit TOM40 and Mitochondrial import receptor subunit TOM7

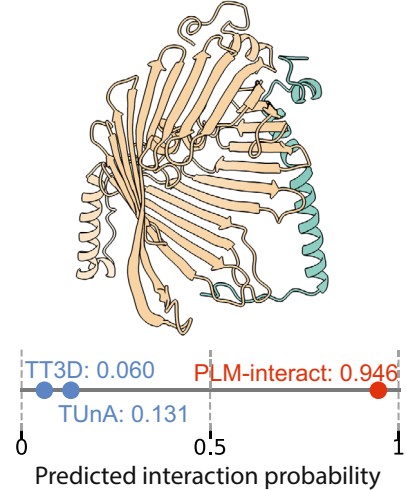

TT3D: 0.060    PLM-interact: 0.946
TUnA: 0.131

0        0.5         1
Predicted interaction probability

## *E. coli*
### ABC transporter permease protein and Possible ABC-transport protein, ATP-binding component

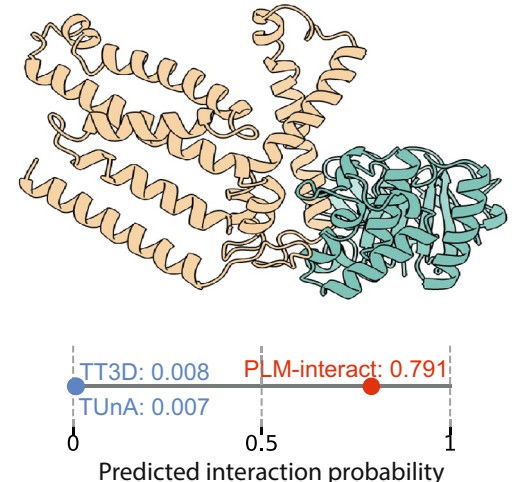

TT3D: 0.008    PLM-interact: 0.791
TUnA: 0.007

0        0.5         1
Predicted interaction probability

**Fig. 3 | Protein-protein interaction (PPI) example for each species that is predicted correctly by PLM-interact but not by TUnA and TT3D.** Protein-protein structures are predicted by Chai-1[28] and visualised with ChimeraX[45]. The predicted interaction probabilities of PPI models range between 0 and 1. A predicted interaction probability > 0.5, is considered to be a positive PPI, while < 0.5 is a negative pair. Interacting proteins are shown from left (yellow) to right (green), respectively. **Mouse**: P97287 (Induced myeloid leukaemia cell differentiation protein Mcl-1 homologue) and P63028 (Translationally-controlled tumour protein); **Fly:** Q9W0F0 (Dynein light chain roadblock) and Q7K035 (AT23443p); **Worm:** Q21955 (Mediator of RNA polymerase II transcription subunit 15) and Q9N4F2 (Mediator of RNA polymerase II transcription subunit 19); **Yeast:** P23644 (Mitochondrial import receptor subunit TOM40) and P53507 (Mitochondrial import receptor subunit TOM7); and **E. coli:** A0A454A7G5 (ABC transporter permease protein) and A0A454A7H5 (Possible ABC-transport protein, ATP-binding component). See Supplementary Fig. 4 for the corresponding AlphaFold3[29] predicted structures. The ipTMs of both Chai-1 and AlphaFold3 for each structure are shown in Supplementary Table 2. Source data are provided as a Source Data file.

polymerisation[25], import of protein precursors into mitochondrial[26] and protein transportation[27]. We use Chai-1[28] and AlphaFold3[29] to predict and visualise these interacting protein structures – these visualisations are shown in Fig. 3 and Supplementary Fig. 4, respectively. The self-assessment scores pTM and ipTM from Chai-1 and AlphaFold3 are reported in Supplementary Table 2.

To investigate if the order of protein in each test pair has an impact on the prediction results, we also perform inference on the test protein pairs with the reversed order. We observe almost identical AUPR performance (Supplementary Fig. 5) and predicted interaction probability distributions (Supplementary Fig. 6).

To probe the role of sequence identity on PLM-interact's performance, we evaluate the model's performance with the different protein identities between training and test datasets (see "Methods" for details). As expected, we observe marked sequence similarity between human and mouse, similarities reduce significantly between fly, worm, yeast and E. coli with human. Notably, both PLM-interact and the second-best performer, TUnA, in the cross-species benchmark benefit from sequence similarity. Their performance improves as protein identity increases, with PLM-interact consistently outperforming TUnA (Supplementary Fig. 7).

To further evaluate our model's performance in relation to sequence similarity, we train PLM-interact on a leakage-free human 'gold' standard training dataset created by Bernett et al.[30] and compare with the state-of-the-art PPI approaches. In this benchmarking dataset,

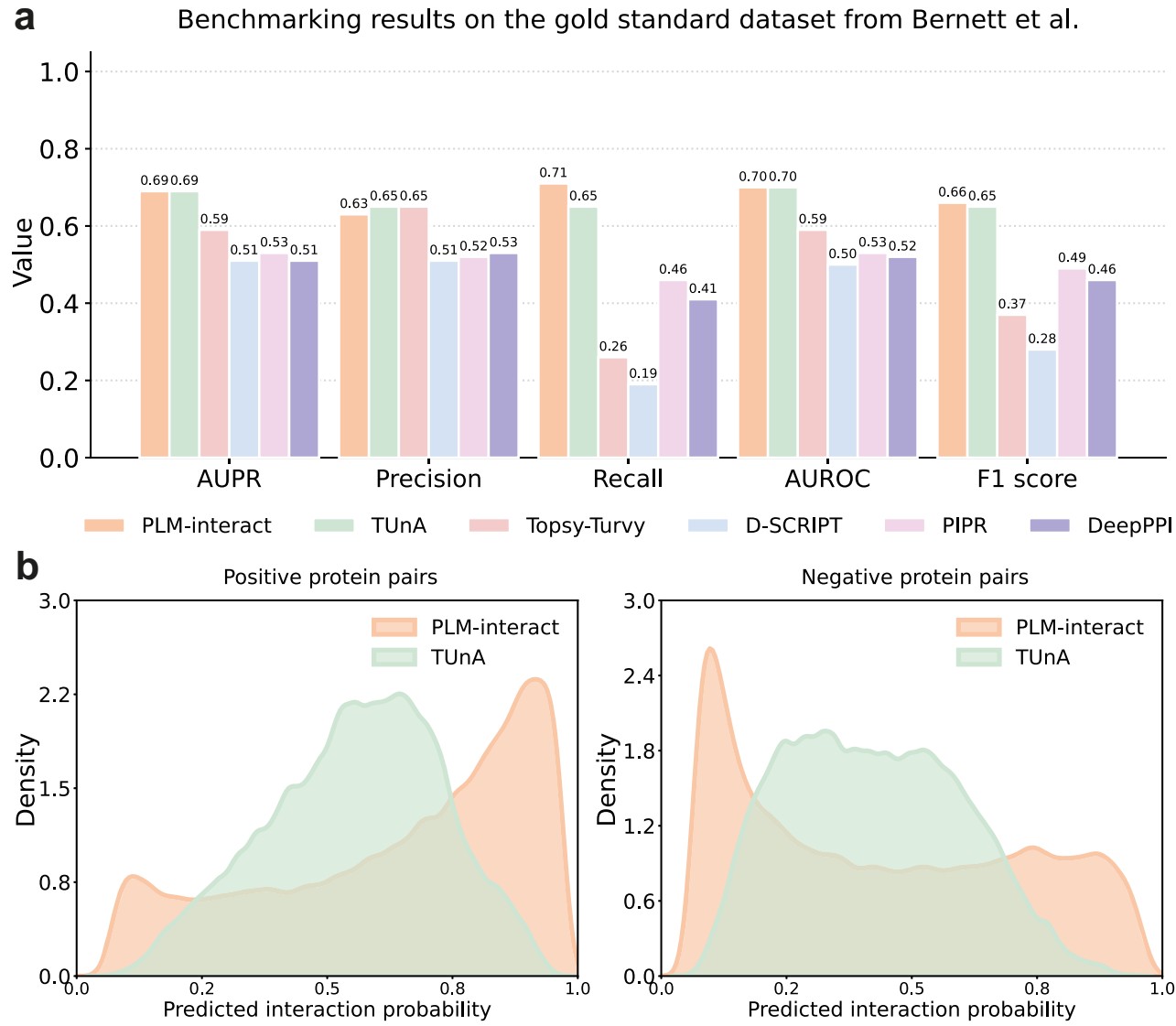

**a** Benchmarking results on the gold standard dataset from Bernett et al.

**Fig. 4 | Performance comparison of protein-protein interaction (PPI) models on the Bernett benchmarking dataset[30]. a** The x-axis shows the evaluation metrics (AUPR, Precision, Recall, AUROC and F1-score) for six PPI models: PLM-interact, TUnA, Topsy-Turvy, D-SCRIPT, PIPR and DeepPPI, the *y*-axis represents the corresponding metric values. **b** The distribution of predicted interaction probabilities of PLM-interact and TUnA for the positive and negative protein pairs, respectively. Source data are provided as a Source Data file.

there are no overlaps and minimal sequence similarities among the training, validation and test datasets.

Due to computing limitations on the maximum sequence length of a pair, we only use 80% of the training set for training. The result on the test set is shown in Fig. 4, where PLM-interact exhibits identical AUPR (0.69) and AUROC (0.7) to TUnA. Interestingly, when adopting a neutral 0.5 threshold on predicted interaction probabilities for final classification, PLM-interact outperforms TUnA and other baselines in F1-score and recall. The improvement in recall is 9% over TUnA, while the precision is comparable with TUnA, indicating PLM-interact performs better at predicting positive interactions.

### Fine-tuned PLM-interact can identify the impact of mutations on interactions

Here, we examine PLM-interact's ability to predict the mutation effect on interactions. We use mutation data from IntAct[31], specifically, mutations that increase (IntAct ID: MI:0382[32]) or decrease (IntAct ID: MI:0119[33]) interaction rate or strength of binding (Fig. 5a). These collectively give us 6,979 total annotated mutation effects. Each annotated mutation effect sample consists of the wild type and mutant sequences of one protein and its interacting protein, which is always in its wild type, i.e., only a single protein is mutated in a PPI. The effect is treated as a binary label for increasing (+) or decreasing interaction (-). To make predictions on the mutation effect, we compute the log-predicted interaction probability ratio between the mutant and canonical (wild type) pairs. A positive log ratio indicates the increasing interaction class and negative otherwise (Fig. 5b). This strategy is similar to the log-likelihood ratio approach for variant effect prediction in single proteins[34,35].

We develop a fine-tuning strategy for predicting mutation effect on interactions (see Fig. 5b and "Methods"). Our method leverages the log-predicted interaction probability ratio as input to a cross-entropy loss, allowing the gradient to be backpropagated to update all layers of PLM-interact. This training allows the model to calibrate changes in predicted interaction probability in mutant case in relation to the canonical scenario.

## a Binary mutation effect classification task

## b Inference and fine-tuning for mutation effect prediction

**Fig. 5 | Predicting mutation effects on protein-protein interactions (PPIs).**
**a** Diagrammatic overview of the binary mutation effect classification task.
**b** Inference and fine-tuning of PLM-interact to predict mutation effects that increase or decrease interaction rate/strength. The log formula in this panel is the log-predicted interaction probability ratio between the mutant and canonical pairs.

**c** Precision-Recall curves of two fine-tuned PLM-interact models and four zero-shot models (PLM-interact, TUnA, Topsy-Turvy and D-SCRIPT) on the test dataset. **d** The ROC curves of the fine-tuned PLM-interact models and four zero-shot models on the test dataset. Source data are provided as a Source Data file.

We construct training, validation and testing sets from the combined IntAct increasing and decreasing interaction set (see Datasets for details) and benchmarked two fine-tuned PLM-interact and four models without fine-tuning (i.e., zero-shot), including TUnA, Topsy-Turvy, D-SCRIPT, and a zero-shot PLM-interact (Fig. 5c, d). We find all zero-shot models to perform poorly, with close to random performance in AUPR and AUROC, despite the overlapping proteins existing between the human PPI training set and the mutation PPI test set. Remarkably, we observe marked improvement (150% in AUPR and 36% AUROC) when fine-tuning all layers of PLM-interact. To assess how traditional methods perform in the fine-tuning setting, we also train a

version of PLM-interact where only the classification head is fine-tuned; the result is significantly worse than fine-tuning the full model.

We show an example of a successfully predicted mutation that increases interaction strength or rate in Fig. 6a, the DNA replication licensing factor MCM7, which is important for DNA replication in human cells. MCM7 is reported as a biomarker in human cancers such as hepatocellular carcinoma and lung cancer[36,37]. MCM7 Y600 phosphorylation is associated with breast cancer, and the mutant Y600E of MCM7 enhances the interaction levels with MCM members[38]. PLM-interact predicts a positive log ratio of 0.165 between the mutant PPI and the canonical PPI, correctly suggesting an increase in interaction.

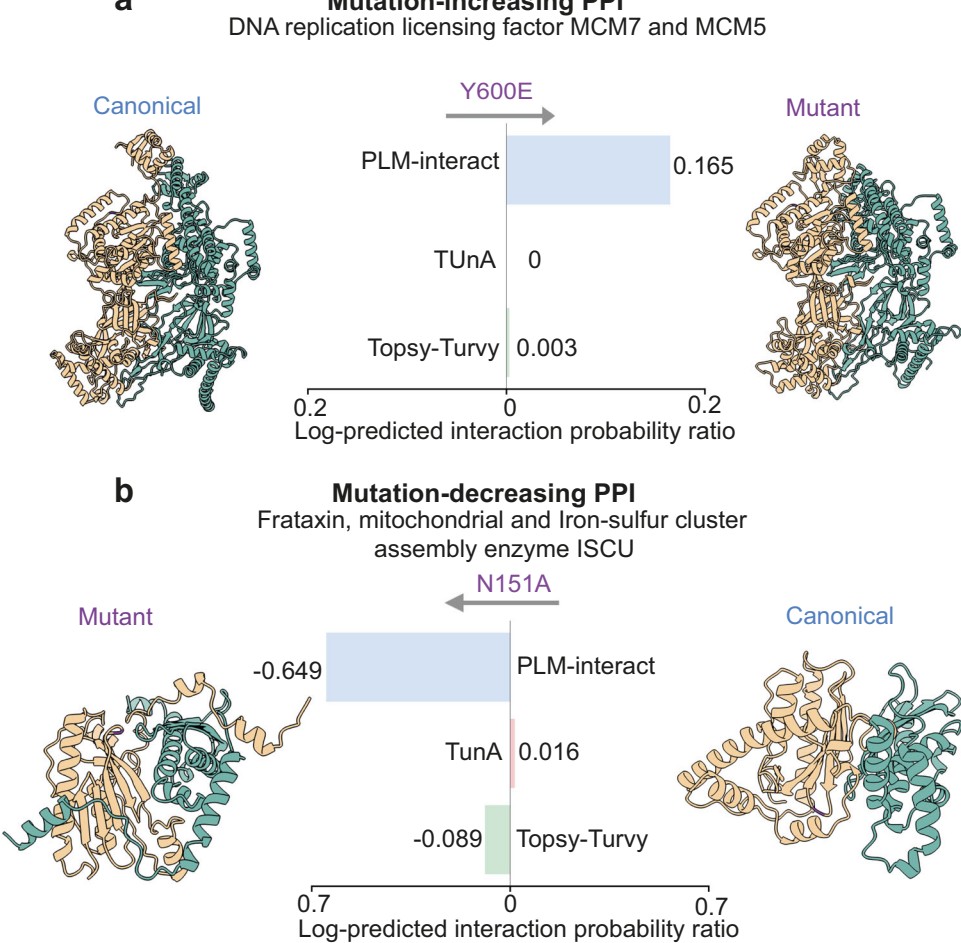

**Fig. 6 | Demonstration of PLM-interact detecting changes in human protein-protein interactions (PPIs) associated with mutations. a** shows an example of a mutation causing an increase in binding affinity, while (**b**) shows a mutation causing a decrease in binding affinity. These PPI structures are predicted using Chai-1[28] and visualised with ChimeraX[45]; here, the mutated amino acids are highlighted in purple. In each panel, the log-predicted interaction probability ratio between the mutant and canonical protein pairs is shown for fine-tuned PLM-interact and zero-shot TUnA and Topsy-Turvy, respectively. The positive log ratio indicates a mutation-increasing PPI, while the negative log ratio indicates a mutation-decreasing PPI. The ipTMs of both Chai-1 and AlphaFold3 for each structure are shown in Supplementary Table 3. Interacting protein structures are shown from left (yellow) to right (green). **a** Residue 600 Tyrosine (Y) of P33993 (DNA replication licensing factor MCM7) is mutated to Glutamic Acid (E), increasing its interaction with P33992 (DNA replication licensing factor MCM5). **b** Residue 151 Asparagine (N) of Q16595 (Frataxin, mitochondrial) is mutated to Alanine (A), decreasing interaction with Q9H1K1 (Iron-sulfur cluster assembly enzyme ISCU). See Supplementary Fig. 8 for the corresponding AlphaFold3 predicted structures. Source data are provided as a Source Data file.

Next, we show an example of the mutation that decreases interaction (Fig. 6b). The protein Frataxin encoded by FXN is important for the synthesis of iron–sulfur cluster, and a mutation of FXN has been discovered to be associated with a neurodegenerative disease Friedreich's ataxia (FRDA)[39]. The wild-type FXN interacts with ISCU (Iron-sulfur cluster assembly enzyme ISCU). The N151A variant of FXN is reported to decrease the binding affinity with ISCU[40]. PLM-interact correctly predicts that the missense mutation N151A reduces the interaction probability (log ratio is − 0.649).

### Improved virus-human PPI prediction
To study virus-host PPI prediction, we train PLM-interact on a virus-human PPIs dataset from Tsukiyama et al.[11]. The dataset is derived from the Host-Pathogen Interaction Database (HPIDB) 3.0[41], and comprises a total of 22,383 PPIs, which include 5882 human and 996 virus proteins. We compare our model with three recent virus-human PPI models: PLM-based approach STEP[14] and the protein embeddings-based approaches LSTM-PHV[11] and InterSPPI[42]. STEP is similar to

existing PPI models benchmarked previously in our study; it leverages protein sequence embedding extracted by the pre-trained PLM ProtBERT[43]. The results show that PLM-interact outperforms the other models. For the STEP comparison, this corresponds to improvements in AUPR, F1 and MCC scores of 5.7%, 10.9% and 11.9%, respectively (Fig. 7a). The length of virus, human proteins and the combined length of virus-human PPIs are shown in Fig. 7b. To further analyse our model's performance, we select three pairs of virus-human PPIs from our test data, all with corresponding experimental virus-human complex structures available in the HVIDB[44]. We then use ChimeraX[45] to visualise these structures and present PLM-interact's predicted interaction probability for each example (see Fig. 7c).

### Discussion
In this study, we have developed PLM-interact, a PPI predictor that extends single protein-focused PLMs to their interacting protein partner. We report significant improvements in held-out species comparisons and with further fine-tuning highlight successful

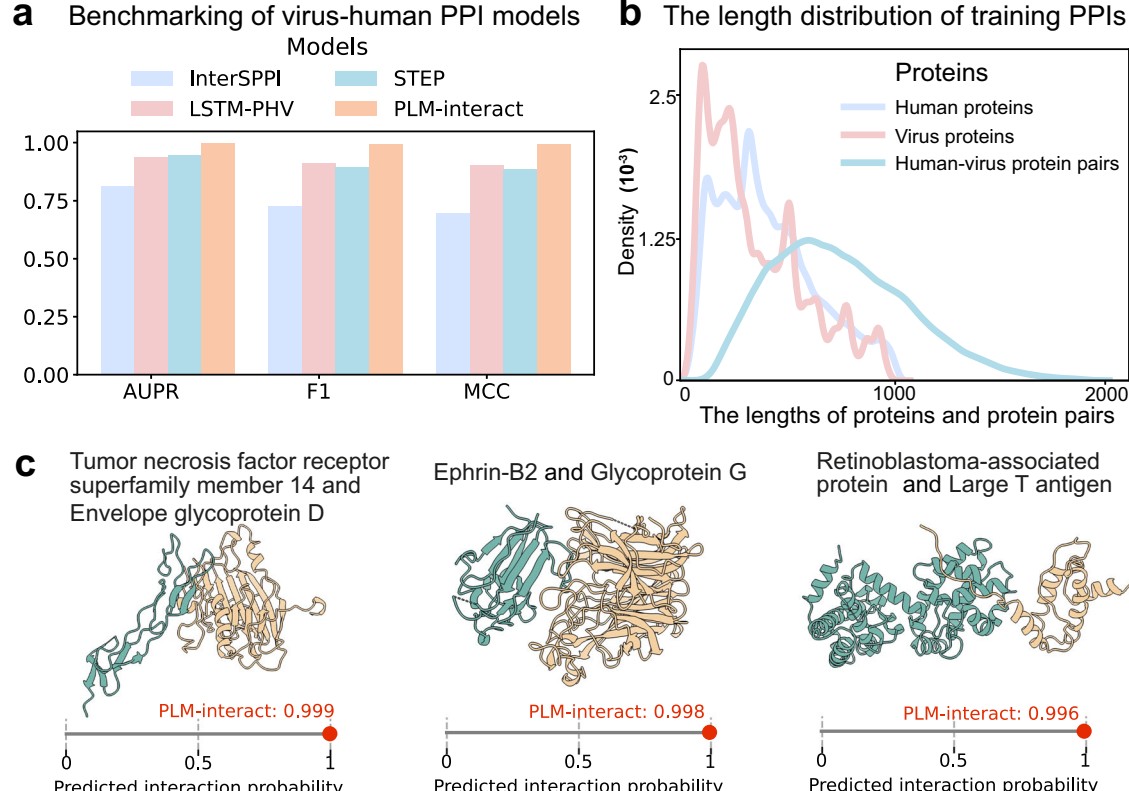

**Fig. 7 | The benchmarking results of virus-human protein-protein interaction (PPI) models. a** Comparison of AUPR, F1 and MCC metrics of PLM-interact against recent virus-human PPI models. **b** The distribution of the length of virus proteins, human proteins and virus-human protein pairs. **c** The virus-human PPIs shown are correctly predicted by our model, and their 3D complex structures are experimentally verified and obtained from the human-virus PPI database (**HVIDB**[44]). From left (green) to right (yellow), these interacting protein structures are: Tumour necrosis factor receptor superfamily member 14 (Human protein: Q92956) with Envelope glycoprotein D (human herpes simplex virus 1: P57083), Ephrin-B2 (human protein: P52799) with Glycoprotein G (Nipah virus protein: Q9IH62) and Retinoblastoma-associated protein (human protein: P06400) with Large T antigen (Simian virus 40: P03070). Note: The metrics results of the other three models in panel (**a**) are taken from STEP[14] paper. Source data are provided as a Source Data file.

examples of predicting mutational effects on protein interactions. We further demonstrate PLM-interact's performance in a virus-human PPI prediction task, showing a significant improvement over state-of-the-art prediction approaches.

Underlying the benefit of PLM-interact is the improved capability of correctly predicting positive PPIs in the held-out species. Notably, PLM-interact, based solely on a large language model approach, significantly outperform two baselines that incorporate multi-modal input, namely TT3D[13] and Topsy-Turvy[20]. TT3D includes explicit structural information, the per-residual structural alphabet from Foldseek[46]. Topsy-Turvy incorporates network data. Inclusion of such additional features should further improve PLM-interact's performance.

Furthermore, our fine-tuning experiments show the potential of predicting mutation effects on PPIs from sequence alone. This could lead to interaction-aware in-silico variant effect predictors where methods rely on PLMs of the single proteins[34,47,48]. However, current training data remains limited. The number of high-quality structures of mutant proteins and their interaction partners are low. Algorithmically, models with long and multimodal context[49–51] that include multiple proteins, structures and nucleotides could be specialised for interaction tasks.

Finally, effective sequence-based virus-host PPI predictors could provide the much-needed molecular detail to conventional virus-host species prediction tools, which tend to rely on genome composition signals, ignoring host molecules that are interacting physically with

viral molecules[52–55]. In those approaches, the host species only acts as a label. Recent progress within SARS-CoV-2 PPI studies mapped out a complex interaction landscape between the virus and human proteome[56,57]. Other viruses are likely to have similarly complex interactions with human and animal hosts. Leveraging these interactions could lead to tools that are better at predicting zoonotic events and the potential for the emergence of novel viruses. While PLM-interact has demonstrated significant improvements, there is much to do in terms of generating reliable predictions, in particular, the need for high-quality virus-host experimental PPI data for training. What is clear is that attention-based large language models applied to longer-range sequence interactions are enhancing our understanding of both proteins and their interactions--the fundamental 'language' of molecular biology.

## Methods

### Datasets
Overview of all datasets used in this paper:

1. **Cross-species dataset:** The human training and five test datasets: mouse, fly, worm, yeast and *E. coli* from Sledzieski et al.[15] are used for benchmarking between PLM-interact and the state-of-the-art PPI approaches (Fig. 2), and PPI model evaluation under the different protein sequence identity between training and test datasets (Supplementary Fig. 7).
2. **Bernett dataset:** The leakage-free human training, validation and test benchmarking dataset created by Bernett et al.[30] is

used to further benchmark PLM-interact and other PPI approaches (Fig. 4).

3. **Mutation effect dataset:** The mutation dataset collected from IntAct[31] is used to fine-tune PLM-interact for predicting increasing or decreasing strengths of interactions associated with mutations (Figs. 5 and 6).

4. **Virus-human PPI dataset:** The virus-human benchmarking dataset from Tsukiyama et al.[11] is used to compare PLM-interact and other virus-human PPI models (Fig. 7).

5. **STRING V12 training dataset:** The database STRING V12[58] is used to construct a larger dataset for PLM-interact training.

## Cross-species dataset

The benchmarking human PPI dataset, from Sledzieski et al.[15], comprises human training and validation data, and test data from five other species: mouse, *Mus musculus*; fly, *Drosophila melanogaster*; worm, *Caenorhabditis elegans*; yeast, *Saccharomyces cerevisiae*; and *E. coli*, *Escherichia coli*, all retrieved from STRING V11[59]. We train and validate our model on human PPIs and then conduct inference on PPIs from five other species. All training, validation and test datasets maintain a 1:10 ratio of positive to negative pairs, reflecting the fact that positive PPIs are significantly fewer than negative pairs in PPI networks. Negative pairs are generated by randomly pairing proteins not reported to interact. The length of protein sequences ranges from 50 to 800, and PPIs are clustered at 40% identity using CD-HIT[60] to remove the redundant PPIs. The human training dataset includes 38,344 positive PPIs, whereas the validation set includes 4794 positive PPIs. Each of the five species includes 5000 positive interactions, except for *E. coli*, which only has 2000 positive interactions due to the fewer positive PPIs in the STRING dataset used[15].

## Bernett dataset

The human benchmarking dataset constructed by Bernett et al. explicitly to minimise data leakage, is also used to evaluate PPI models' performance[30]. The positive protein pairs are from the HIPPIE v2.3[61] human PPI dataset, and negative protein pairs are randomly selected, and the ratio between positive and negative pairs is 1:1. Their data-splitting strategy removes overlaps and minimises sequence similarity among the training, validation and test datasets. There are 163,192 training protein pairs, 59,260 validation protein pairs and 52,048 test protein pairs. CD-HIT[60] was used to remove redundant protein pairs that share more than 40% protein sequence similarity with existing training protein pairs. Due to the storage limitation, we train on protein pairs that have the maximum combined paired length of 2193, which covers 80% of this benchmarking training data.

## Mutation effect dataset

The mutation effect dataset is obtained from IntAct[31]. We collect mutations that increase (MI: 0382) or decrease (MI: 0119) interaction rate or strength. Each sample is a group of three proteins consisting of a canonical protein, a mutant protein and a participant protein. We remove PPIs that have the same canonical and participant proteins, i.e., self-interactions. The resulting dataset contains 1281 mutation-increasing PPI samples and 5698 mutation-decreasing PPI samples (Supplementary Fig. 9).

Due to GPU memory limitations on the length of protein pairs, we prioritise short protein pairs (< 2201 amino acids, ~80% of the data) to maximise the training size, while allowing the validation and test sets to have pairs with much longer combined lengths. In the end, we have 5103 training, 841 validation and 841 test samples.

## Virus-human PPI dataset

The benchmarking dataset of 22,383 virus-human PPIs includes 5882 human and 996 virus proteins. This dataset was obtained from Tsukiyama et al.[11], sourced from the HPIDB 3.0 database[41]; the ratio of positive to negative pairs is 1:10 and negative pairs are chosen based on sequence dissimilarities. The length of protein sequences ranges from 30 to 1000, and the redundant PPIs are filtered based on a threshold of 95% identity using CD-HIT[60]. The processed dataset was split into training and test datasets with a ratio of 8:2. Our training and test split is identical to the one in Tsukiyama et al.[11].

## STRING V12 training dataset

In addition, we provide a model trained on human PPIs from STRING V12[58]. The positive PPIs are selected by collecting physical links with positive experimental scores, while excluding PPIs with positive homology scores and confidence scores below 400. Previous studies have typically limited the maximum length of protein sequences to 800 or 1000 due to GPU memory limitations. We process the training protein sequences with a combined length threshold for protein pairs of 2101. This human dataset includes 60,308 positive PPIs for training and 15,124 positive PPIs for testing. Furthermore, protein sequences are clustered at 40% identity using MMSeq2[62], and only PPIs from the distinct clusters are chosen to eliminate redundant PPIs. Again, the positive-to-negative protein pair ratio is 1:10, consistent with the aforementioned two benchmarking datasets.

## Model architecture

We use ESM-2 as the base model in PLM-interact. ESM-2 is an encoder transformer model with a parameter size range from 8 million to 15 billion. The results presented are PLM-interact based on ESM-2 with 650 M parameters. We also provide PLM-interact model checkpoints trained with ESM-2 35 M on our Hugging Face repository to help with testing. The input representation contains amino acid token representations from two proteins. This setup is similar to the original BERT model[63], also known as the cross-encoder, which simultaneously encodes a pair of query and answer sentences.

A standard input sequence of PLM-interact, $x$, can be shown as the following:

$$x = [CLS, P_1, EOS, P_2, EOS], \qquad (1)$$

where $CLS$ is the classification token, $P_1$ contains amino acid tokens of protein 1, $P_2$ contains amino acid tokens of protein 2, and $EOS$ is the end-of-sentence token. The initial EOS token marks the end of the amino acid sequence in protein 1. This setting allows us to use the original ESM-2 tokenizer to generate embedding vectors $e$, and pass them to the transformer encoder of the ESM-2:

$$h = f(e), \qquad (2)$$

where $f$ is ESM-2, $e$ contains the token embeddings of $x$, and $h$ contains the output embeddings of all input tokens. $h$ can be presented as:

$$h = \left\{ h_{cls}, h_{a_1}, \ldots h_{EOS}, \ldots h_{a_n}, \ldots h_{EOS} \right\}, \qquad (3)$$

where $h_{a_1}$ and $h_{a_n}$ represent amino acid tokens in proteins 1 and 2. Then, we use the $CLS$ token embedding to aggregate the representation of the entire sequence pair and as the features for a linear classification function $\varphi$, and parameterised as a single feed-forward layer with a ReLU activation function. The output of the FF layer is converted by the sigmoid function $\sigma$ to obtain the predicted interaction probability $g$,

$$g = \sigma(\varphi(h_{cls})). \qquad (4)$$

## Model training

PLM-interact is trained with two tasks: (1) a mask language modelling (MLM) task predicting randomly masked amino acids and (2) a binary

classification task predicting the interaction label of a pair of proteins. PLM-interact is trained for 10 epochs using a batch size of 128 on both benchmarking datasets of human PPIs and virus-human PPIs. For all training runs, the input protein pairs are trained using both orders as the interaction between protein 1 and protein 2 is the same as the protein 2 and protein 1, which leads to doubling of the training set size. The validation and testing sets are not subject to the same data argumentation. The learning rate is 2e-5, weight decay is 0.01, warm-up is 2000 steps, and the scheduler is WarmupLinear, which linearly increases the learning rate over the warm-up steps. These parameters are the same as the cross-encoder training in the Sentence-BERT paper[18]. During training, we evaluate the model's performance at every 2000 steps on the validation set. For every evaluation, a set of 128 protein pairs are randomly sampled from the validation set, and the results are averaged over 100 times to ensure metric reliability. Here, we use both masking and classification losses to optimise our model, the loss function for each data point $l$ can be represented as:

$$l = \alpha l_{mlm} + \beta l_{ce}, \tag{5}$$

where $l_{mlm}$ and $l_{ce}$ are separately represent the MLM loss and classification (i.e., cross entropy) loss. $l$ can be written as:

$$l = -\frac{\alpha}{M}\sum_{i=1}^{M} \ln p(x_i|x_{-i}) - \beta(y\ln(g) + (1-y)\ln(1-g)), \tag{6}$$

where $M$ is the number of the masked tokens, $x_i$ is the true token at position $i$, $p(x_i|x_{-i})$ is the probability of the true token $x_i$ given the unmasked amino acid $x_{-i}$. $y$ is the label of the interaction, and $g$ is the predicted probability for $y = 1$, obtained from Eq. 4. $\alpha$ and $\beta$ are weights for the MLM and classification losses, and they are determined in the following "Technical benchmark for hyperparameter selection" section.

All of the models are trained on the DiRAC Extreme Scaling GPU cluster Tursa. A typical 10-epoch training run of the model with ESM-2 (650 M) with human PPIs takes 31.1 h on 16 A100-80 GPUs. A typical 10-epoch training run of the model with ESM-2 (650 M) trained on virus-human PPIs takes 30.5 h on 8 A100-80 GPUs. The model with ESM-2 (650 M) trained on STRING V12 human PPIs used 16 A100-80 GPUs for 86.4 h. For model training time with different ratios and model sizes, see the following section, technical benchmark and Supplementary Table 1 for details.

We provide model checkpoints that include human PPI models trained on the benchmarking dataset constructed by Sledzieski et al.[15] and retrieved from STRING V11[59], a human PPI model trained on a benchmarking dataset created by Bernett et al.[30] and sourced from HIPPIE v2.3[61], a virus-human PPI model trained on the benchmarking virus-human PPIs created by Tsukiyama et al.[11] and sourced from HPIDB 3.0[41], as well as a human PPI model trained on human PPIs that we collected from STRING V12[58].

### Technical benchmark for hyperparameter selection
To find the optimal value of $\alpha$ and $\beta$ in Eq. (6), we benchmark a range of different options between mask loss and classification loss on human benchmarking data. For each ESM-2-35M and ESM-2-650M model, we train five models with different settings of ratios $\alpha$: $\beta$ between mask loss and classification loss. The ratios are $\alpha$: $\beta = 1{:}1$, 1:5, 1:10, 0:1 (with mask), and 0:1 (without mask, denoted as classification) (Supplementary Fig. 1a). The difference between 0:1 (with mask) and 0:1 (without mask) is whether masking the training protein pairs before inputting the CLS token embedding to the classification layer. We used the human validation set for each model to identify the optimal epoch checkpoint achieving the best AUPR. Next, the final model is selected based on testing on five other host PPIs.

PLM-interact trained with ESM-2-650M performs better than models trained with ESM-2-35M (Supplementary Fig. 1b, c). We find that a ratio of 1:10 is the optimal choice for ESM-2-650M. The AUPR of *E. coli* shows a 4.3% improvement over the second-best model, while other species remain comparable or better with this 1:10 ratio (Supplementary Fig. 1c). According to these results, we select a loss ratio of 1:10 for ESM-2-650M. The ratio setting is implemented in benchmarking of human PPI training and virus-human PPI training, as well as human PPI training using the STRING V12 database.

### Performance with different masking percentages
Typically, 15% tokens are masked out to train protein language models, such as ESM-1b[47] and ESM-2[17]. Given that PLM-interact deals with much longer sequence lengths than typical ESM-2, we test the model's performance under different masking percentages. We test a range of different masking percentages (7%, 15%, 22% and 30%) to train ESM-2-650M with a mask-to-classification loss ratio of 1:10.

We also included a binary model without masking as the base model. We compare this base model with masking models that are trained with different masking ratios to determine the benefit of masking percentages on performance. We use McNemar's test to evaluate if the masking model is significantly better than the binary model ($p$-value < 0.05). As this is an unbalanced binary classification task, we determine true and false predictions using the threshold that gives the best F1 score. Finally, we compare the AUPR performance of each model on five test species.

The results are shown in Supplementary Fig. 2. For each species, we show a line plot of AUPR performance with the different masking ratios: 0% (binary), 7%, 15%, 22% and 30%. A gold star is marked if the masking model is significantly better than the binary classification model (0% masking). The model with 15% masking ratios is the only masking model that consistently outperforms the binary model. Overall, we also observe insignificant differences among the masking percentages.

### Inference in the binary mutation effect task
Let $x_{mutant}$, $x_{canonical}$, and $o_{effect}$ be the mutant, canonical interaction pair representation and binary effect label, respectively. The basic data unit in this task contains a triplet $\{x_{mutant}, x_{canonical}, o_{effect}\}$.

Inference is performed based on computing the log-predicted interaction probability ratio between the mutant and canonical protein pairs:

$$lr = \log\left(\frac{g_{mutant}}{g_{canonical}}\right) \tag{7}$$

Where $g$ is the predicted interaction probability in Eq. 4. A positive log ratio indicates a positive class and negative otherwise. Both AUPR and AUROC are computed based on the log ratio itself.

### Fine-tuning in the binary mutation effect task
We use a binary cross-entropy loss to fine-tune PLM-interact as follows:

$$l_{FT} = -(o_{effect}\ln(\sigma(lr)) + (1 - o_{effect})\ln(1 - \sigma(lr))) \tag{8}$$

Here, $\sigma$ is the sigmoid function to obtain the predicted interaction probability of the binary mutation effect classification. We fine-tune two versions of PLM-interact models, one with all layers fine-tuned, the other with only the classification layer being updated. The latter is designed to mimic fine-tuning the traditional approach of a frozen pretrained protein encoder followed by a learnable classification head.

We use the checkpoint trained with the human dataset from Sledzieski et al.[15] for two versions of PLM-interact fine-tuning. Both fine-tuned models are trained for 40 epochs, and the best epoch is obtained when the validation loss reaches its minimum

(Supplementary Fig. 10). The effective batch size is 128, the learning rate is 2e-5, weight decay is 0.01, warm up is 2000 steps and the schedular is WarmupLinear, which linearly increases the learning rate over the warmup steps.

## Baselines

We compute the prediction interaction probabilities based on checkpoints of TUnA[19], TT3D[13], Topsy-Turvy[20] and D-SCRIPT[15] to generate precision-recall (PR) curves in Fig. 2b. Due to the absence of publicly available checkpoints for DeepPPI[21] and PIPR[6], these methods are excluded from the PR curve comparison. The AUPR value for TUnA is sourced from the TUnA paper, the AUPR values for DeepPPI and Topsy-Turvy are sourced from the Topsy-Turvy paper[20], those for D-SCRIPT and PIPR are from the D-SCRIPT paper[15], and the AUPR value of TT3D is obtained through email communication. For the benchmarking on the Bernett dataset[30], we obtain the AUPR value of TUnA from the TUnA paper[19] and other models' AUPR values are obtained from Bernett et al.[30]. A complete list of the main features, architectures, references and code links for each baseline method can be found in Supplementary Table 4. As for model comparison on the mutation effect prediction task, Topsy-Turvy and D-SCRIPT cannot handle protein pairs longer than 2000, resulting in 598 out of 841 test samples being used for prediction. Therefore, we show Precision-Recall and ROC curves under these 598 predictions in Fig. 5c, d.

## MMseq2

We use the sequence search and clustering tool MMseq2 Release 13-45111[62] to obtain the protein sequence-based alignment results between each pair of proteins; the parameters setting is: --threads 128 --min-seq-id 0.4 --alignment-mode 3 --cov-mode 1.

## Protein sequence similarities between training and test datasets

To evaluate each PPI model's performance with different levels of protein sequence similarities between training and test datasets, we use MMseq2[62] to obtain the protein sequence identity for each protein in the test pairs from five test species against all human training proteins, with identity values ranging from 0 to 100. The sequence identity between each test protein pair and training proteins is determined by the maximum identity of any protein in each test pair. In Supplementary Fig. 7, we report the AUPR values of PLM-interact and TUnA under different levels of identity "bins" [0, 20, 40, 60, 80, 100] between training and test proteins.

## Chai-1

Chai-1[28] is a state-of-the-art model for molecular structure prediction, available at https://lab.chaidiscovery.com/. We use Chai-1 with the "specify restraints" option to predict protein-protein structure complexes and visualise predicted PPI structures using the molecular visualisation programme ChimeraX-1.7.1[45].

## AlphaFold3

AlphaFold3 is a tool to predict the biomolecular interactions, including protein, DNA, small molecules, ions and modified residues[29], available at https://alphafoldserver.com/. We use AlphaFold3 in its PPI mode to predict protein structure complexes. The results are visualised with the molecular visualisation programme ChimeraX-1.7.1[45].

## Self-assessment scores

The ipTMs and pTM scores for the Chai-1[28] and AlphaFold3[29] predicted structures are reported in Supplementary Table 2 and Supplementary Table 3. IPTM scores below 0.6 indicate failed predictions, while scores above 0.8 indicate high confidence predictions. pTM scores above 0.5 suggest that the predicted structures are similar to the ground truth. For more details about these self-assessment scores, refer to the AlphaFold3[29] paper and official description at https://alphafoldserver. com/faq#how-can-i-interpret-confidence-metrics-to-check-the-accuracy-of-structures.

## McNemar's test

McNemar's test[64] is a statistical test that determines if there are significant differences between paired nominal data.

$$McNemar's test = \frac{(b - c)^2}{(b + c)} \qquad (9)$$

Here, $b$ represents the count of correct predictions obtained by $model1$ and incorrect predictions by $model2$, while $c$ represents the count of incorrect predictions obtained by $model1$ and correct predictions by $model2$.

To investigate if our models with masking perform significantly differently from a binary model without masking, we conducted a McNemar's test for any models under different masking percentages with the binary classification model (0% masking). This test is based on the number of correct and incorrect between two models. Predicted interaction probabilities from each model are used to get predicted labels, which are used to obtain the counts of correct and incorrect predictions. A McNemar's test $p$-value < 0.05 indicates a significant difference between the predictive performance of two models. The model with more correct predictions is considered superior to the other.

## Reporting summary

Further information on research design is available in the Nature Portfolio Reporting Summary linked to this article.

## Data availability

The cross-species benchmarking dataset created by Sledzieski et al.[15] is available at https://d-script.readthedocs.io/en/stable/data.html. The human dataset created by Bernett et al.[30] is available at https://doi.org/10.6084/m9.figshare.21591618.v3. The virus-human benchmarking PPI dataset created by Tsukiyama et al.[11] is available at http://kurata35.bio.kyutech.ac.jp/LSTM-PHV/download_page. Protein sequences are retrieved from UniProt[66] (https://www.uniprot.org/). The 3D complex structures of human-virus PPIs are obtained from HVIDB[44] (http://zzdlab.com/hvidb/download.php). The STRING V12 human training data are sourced from the STRING PPI database V12[58] (https://stringdb-downloads.org/download/protein.physical.links.v12.0.txt.gz) and are available at https://huggingface.co/datasets/danliu1226/STRING_V12_TrainingSet. The training, validation and test datasets for the mutation effect classification task are sourced from the IntAct molecular interaction database[31] (https://ftp.ebi.ac.uk/pub/databases/intact/current/various/mutations.tsv) and are available at https://huggingface.co/datasets/danliu1226/Mutation_effect_dataset. The data generated in this study (training, validation and test datasets used for PPI benchmarking and mutation effect prediction tasks, cross-species PPI model checkpoints; the Bernett PPI model checkpoint; mutation effect classification model checkpoint; predicted interaction probabilities for PPI models on benchmarking tasks; prediction results for mutation effect classification and PPI model evaluations under varying protein sequence identities between training and test datasets) are available at Hugging face (https://huggingface.co/danliu1226) and the Source Data file. All datasets in this study are publicly accessible without restrictions. Source data are provided in this paper.

## Code availability

The code in this study is publicly available and has been deposited in GitHub at https://github.com/liudan111/PLM-interact, under the MIT license. The specific version of the code associated with this publication is archived in Zenodo and is accessible via https://doi.org/10.5281/zenodo.16643324[65]. Trained model checkpoints and datasets used in this study are available at https://huggingface.co/danliu1226.

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

## Acknowledgements

D.L.R. acknowledges funding from the European Union's Horizon 2020 research and innovation 562 programme, under the Marie Sklodowska-Curie Actions Innovative Training Networks 563 grant agreement no. 955974 (VIROINF) for D.L.; a UK Medical Research Council (MRC) Doctoral Training Programme in Precision Medicine studentship (MR/N013166/1) for K.D.L.; and MRC grants: MC_UU_00034/5, MC_UU_00034/6 and MR/V01157X/1. K.Y. acknowledges support from Cancer Research UK (EDDPGM-Nov21\100001 and DRCMDP-Nov23/100010), Biotechnology and Biological Sciences Research Council (BBSRC) BB/V016067/1, Prostate Cancer UK MA-TIA22-001 and EU Horizon 2020 grant ID 101016851. C.J.M. and A.P. acknowledge support from Cancer Research UK core funding to the CRUK Scotland Institute (A31287) and a core programme award to C.J.M. (A29801). K.Y. acknowledges support from Cancer Research UK core funding to the CRUK Scotland Institute (A31287). This work used the DiRAC Extreme Scaling service (Tursa) at the University of Edinburgh, managed by the Edinburgh Parallel Computing Centre on behalf of the STFC DiRAC HPC Facility (www.dirac.ac.uk). The DiRAC service at Edinburgh was funded by BEIS, UKRI and STFC capital funding and STFC operations grants. DiRAC is part of the UKRI Digital Research Infrastructure.

## Author contributions

D.L. designed the experiments, collected datasets, trained models, wrote the code and prepared the manuscript. F.Y. and K.D.L. contributed to the analysis of the experiments and provided feedback on the experimental design. A.C.Q. provided suggestions for model training and implementing models on the HPC system. A.P. and C.J.M. contributed to the analysis of the experiments. C.M., D.L.R., and K.Y. conceptualised the study, designed the experiments, edited the manuscript, and jointly supervised the research.

## Competing interests

The authors declare no competing interests.
