## [Transparent Peer Review file · Nature Communications]

PLM-interact: extending protein language models to predict protein-protein interactions

Corresponding Author: Dr Ke Yuan

Version 0:

Reviewer comments:

Reviewer #1

(Remarks to the Author)

In this work, the authors extended and fine-tuned the ESM-2 with redesigned input sequences to better capture the interaction information of PPIs. They trained PLM-interact on human PPI datasets and tested the prediction performance on five other datasets of different species. They also validated the applicability of PLM-interact to identify the impact of mutations on interactions. While potentially interesting, I have the following concerns:

1. The PPI datasets were downloaded from the STRING V11 database. The authors should specify the type of these interactions, e.g. physical interactions, functional interactions and how different types might influence the model performance. In addition, is it possible that the datasets contain false positives and false negatives? If so, how to eliminate their influences?
2. The design of the input sentence for PLM-interact is simple and direct. However, a major concern is that the number of proteins currently available is stable and thus a pretrained language model can be used to extract features in advance. For PLM-interact, given new pairs of proteins, the model needs to be retrained, which limits its practical usage.
3. While the prediction performance seems to be astonishing, the baseline models selected are somewhat outdated and thus it is difficult to accurately evaluate the contribution of the PLM-interact.
4. The authors should conduct comparisons by using commonly used classifiers with the learned embeddings.

(Remarks on code availability)

Reviewer #2

(Remarks to the Author)

The authors proposed a protein interaction prediction method, PLMsearch, that extends single protein-focused PLMs (ESM-2) to their interacting protein partner. Below are some questions I hope the authors can address:

1. The author's description of the method is not clear. For example, how exactly did the author fine-tune ESM-2? Specifically, does this process only involve modifying the parameters of the last layer of ESM-2, or were other aspects of the model also adjusted?
2. What is the difference between the Masked Language Modeling (MLM) task used in PLM-Interact and ESM-2? Are there any experimental results demonstrating that performance is optimal when random masking is set at 15%?
3. The article mentions that conventional PLM-based PPI predictor architectures typically include a classification head, but PLM-Interact appears to lack one. Could the authors provide more detailed information regarding the classification-related aspects. In particular, please explain the design of the loss function in more detail.
4. For PPI prediction, many methods use human proteins as the training set and datasets from mouse, fly, worm, yeast, and E. coli as the testing set. However, is there sequence redundancy between these five datasets and the amino acid sequences of human proteins? Could an experiment be conducted to verify whether the model's performance remains strong under different levels of redundancy between the training and testing sets?
5. For virus-human PPI prediction, the authors re-used a virus-human PPIs dataset for model training, which seems to provide a model with different parameters. So, how does the model trained on more than 420,000 human proteins perform in virus-human PPI prediction? How is the generalization ability of the model?
6. Is the experimental purpose clearly defined, particularly in the case of the five positive PPI instances? The self-

assessment score is mainly used to evaluate the accuracy of the predicted structure, rather than to evaluate whether the input sequence has interactions. It seems that the focus is more on explaining the inaccuracies in protein structure predictions from Chai-1 and AlphaFold3 rather than offering new insights into PPI prediction itself.

(Remarks on code availability)

Reviewer #3

(Remarks to the Author)

Overview

The authors introduce PLM-interact, an adaptation of the ESM-2 protein language model fine-tuned with protein-protein interaction (PPI) pairs to predict interactions given only the sequence of each protein. The model is supervised with two objectives, a masked language modeling objective over the paired sequence (separated by EOS tokens) and a classification accuracy (with a small classification head taking a leading CLS token as input), with the balance between these objectives chosen by a hyper-parameter search. They report strong performance compared to previous state-of-the-art methods on a multi-species PPI data set and on a virus-human host PPI benchmark set. They also compare to structure based methods AlphaFold3 and Chai-1, showing sensitivity to gain-of-function and loss-of-function mutations.

Sequence-based protein-protein interaction prediction is an important and well studied problem, especially the application of protein language models thereof. The question of whether language model pre-training can be adapted to include PPIs is well-motivated and the results are compelling. The results on gain- and loss-of-function mutations are especially interesting, since most previous literature has shown limited ability of protein language models to respond to single mutations. The manuscript is overall well-written with a diverse set of experiments. The manuscript contains several valuable insights, such as the analogue to next-sentence prediction in natural language, and the fact that the performance gain of PLM-interact comes largely from the improvement in predicting positive PPIs, rather than improvement in the negative case. The authors develop a new PPI data set based on STRING v12 and release trained models based on this new, large-scale PPI data set. The code base appears clean and well-maintained, with clear training and inference scripts, and pre-trained models are made available through HuggingFace.

The manuscript is currently lacking comparisons with recent state-of-the-art methods and benchmarks, as well as details about how train/test splits were determined that make it difficult to fully assess model performance. Relatedly, several of the biological examples seem arbitrarily picked and the manuscript would benefit from summary statistics, or at least a justification for how the case studies were chosen. Finally, the statistical test used for selection of objective balance hyper-parameters is either fully incorrect or improperly specified, these results likely contain data leakage from the test sets, and the results are not clearly communicated. We provide more details on these and other comments below.

Major Comments

- The authors should evaluate their method on the widely-used gold standard benchmarks from Bernett et al. 2024 ("Cracking the black box..."), the D-SCRIPT data set is a good measure of cross-species performance but is known to have some data leakage from homologous proteins
- The authors should also consider comparing to RAPPID and TuNA, two state-of-the-art sequence-based PPI methods
- The structure-based experiments in Figure 3 are very interesting, but the authors should justify how these examples were chosen apart from others so as to avoid the appearance of cherry picking; or run AlphaFold3/Chai-1 on a wider test set to generate a summary of their performance
- Likewise, the choice of structure-based experiments in Figure 4 should be justified. Especially for mutation sensitivity, the authors should show how the aggregate performance of PLM-interact across the whole InAct data set compares to other sequence-based methods like Topsy-Turvy or PIPR (it would also be good to see AF3/Chai-1, but inference at this scale may not be possible)
- The mutation impact data set contains human proteins, and PLM-interact was trained on human proteins-- the authors should report which proteins in the test set were also seen during training, and report the performance on mutational impact binned by similarity to any protein in the training set. Otherwise, there is a risk that the model is just memorizing true human PPIs and would not generalize to mutations further from this training set
- There is no information provided about how the HPIDB is split into training and test, nor how hyper-parameters were chosen without a validation set. If the splits from the original paper were used (since comparison metrics were taken from that paper), this should be specified
- Since both protein orders (P1 EOS P2 EOS, P2 EOS P1 EOS) were provided at training time but not at inference time, how was the order chosen for validation and testing? Does the model make the same predictions if the order changes? It would be good to provide the model with both orders and average predictions if this value is not identical regardless of order
- It is quite concerning that the values of alpha and beta were chosen based on test set performance with the other species, as this is the final value that is reported-- this is a large source of data leakage, as hyper-parameters should be chosen using a separate validation set. Supp. Fig. 1 shows that the choice of these parameters does not drastically impact performance and the model performance is strong regardless, so this is not disqualifying, but these experiments should really be re-done on the validation set to accurately show the final test-set performance.
- The McNemar's test is an uncommon statistical test that isn't very well motivated, and I believe it is improperly implemented. None of my institutions have access to the original manuscript, but Wikipedia, NE Hawass British Journal of Radiology 2014, and the statsmodels implementation are all in agreement with one another that the formula should be $\frac{2 \cdot (b + c)}{(b + c)^2}$, where b is the count of candidate pairs where model 1 has true positives AND model 2 has false

negatives, and $\$c\$$ is the count of candidate pairs where model 1 has false negatives AND model 2 has true positives. This is not equivalent to what the authors provided, where $\$b\$$ is the `_ratio_` of (model 1 true positives + model 1 true negatives) to (model 2 true positives + model 2 true negatives). As a result, this test statistic is likely not Chi squared-distributed and the p-values are likely incorrect. I would recommend a more common statistical comparison, or the correct implementation of the McNemar's test to compare different models when re-doing model comparison using the validation sets
- Supp Fig 1d, e is very confusing  what does "counts of better model" refer to here? If it is trying to show the relative ranking of each model (which I believe it is) by showing the number of other models it is more significant than, there are much better ways to show this data.

Minor Comments

- Line 52-54: "with a FFN being the dominant option, these classifiers often don't generalize well"  please provide a citation for this unsubstantiated claim
- Lines 124-126: "These PPIs are necessary... protein transportation"  please provide citations for these biological process associations for each protein
- Line 146: How were the thresholds of $ipTM < 0.6$ and $ipTM > 0.8$ chosen?
- Line 324: How were the learning rate, weight decay, etc. chosen?
- Line 338: Please explain that details on selection of alpha and beta are described in a following section
- Line 355: The difference between 0:1 (with mask) and 0:1 (without mask) is confusing-- if this is just whether or not the sequences are masked, why does with with mask task perform better than the pure classification task, which should have more information?

Typographical Errors

- Line 28: add a comma after mutations
- Line 28: add a comma after virology
- Line 56: specify that this context is lacking in model pre-training, not training
- Eq. 7: `model1_false`, not `falae`

(Remarks on code availability)

The code is well-written and well-documented, and the README provides clear instructions for installation of dependencies and the model itself, running training of the model, and performing model inference. The inference is somewhat clunky, as the user is required to define the model themselves; it would be good if the authors slightly cleaned up the inference procedure, but it is clean enough. I was able to download the model checkpoint and run inference. Due to availability of time/compute resources, I did not attempt to re-train the model myself or to replicate validation metrics with the trained model checkpoints.

Version 1:

Reviewer comments:

Reviewer #1

(Remarks to the Author)

The authors have addressed all my concerns.

(Remarks on code availability)

Reviewer #2

(Remarks to the Author)

1、 The authors' description of the methodology is not sufficiently clear. For example, the fine-tuning of the ESM2 model and the use of the self-assessment score should be explained in more detail.

2、 For the virus-human PPI prediction, please provide experimental evidence.

(Remarks on code availability)

The code is available.

Reviewer #3

(Remarks to the Author)

I thank the authors for their thoughtful responses to my comments, which they have suitably addressed in their revisions and in their response to reviews.

(Remarks on code availability)

The code is well-written and well-documented, and the README provides clear instructions for installation of dependencies and the model itself, running training of the model, and performing model inference. The inference is somewhat clunky, as the user is required to define the model themselves; it would be good if the authors slightly cleaned up the inference procedure, but it is clean enough. I was able to download the model checkpoint and run inference. Due to availability of

time/compute resources, I did not attempt to re-train the model myself or to replicate validation metrics with the trained model checkpoints.

Dear Reviewers,

Many thanks for reviewing our manuscript and the insightful comments on our manuscript.

In light of the comments, we have made the following key updates to our paper:

1. **Additional method and dataset for benchmarking:** We added a recently published PPI model, TUnA, and Bennett et al (2024), a dataset that minimises sequence similarity between training, validation and test sets.
2. **Reworked mutation effect prediction study:** We systematically assessed PLM-interact and other baseline methods in a coherent mutation effect dataset from IntAct. We defined a binary mutation effect prediction task where mutations either increase or decrease interactions. We proposed a novel fine-tuning method which achieved significant performance improvements.
3. **The role of sequence similarities between training and test datasets.** Both PLM-interact and the second-best performer in the cross-species benchmark benefit from sequence similarity. Their performance improves as protein identity increases, with PLM-interact consistently outperforming TUnA. We observe marked sequence similarity between human and mouse, similarities reduce significantly between fly, worm, yeast and *E. coli* with human.
4. **The choice of the masking percentage.** We tested a range of different masking percentages (7%, 15%, 22% and 30%) to train and evaluate PLM-interact on the benchmarking dataset created by Sledzieski et al. We observe that masking 15% amino acid for each protein pair during training is optimal.
5. **The influence of the protein order of each test protein pair on PLM-interact's performance.** We demonstrated that performing inference on both the original and reverse order of each test protein pair shows identical distributions of predicted interaction probabilities.

In the following, we reply to each of the reviewer's comments in turn, in blue colour. Please also find the revised manuscript where all changes are marked in yellow.

Reviewer #1 (Remarks to the Author)

In this work, the authors extended and fine-tuned the ESM-2 with redesigned input sequences to better capture the interaction information of PPIs. They trained PLM-interact on human PPI datasets and tested the prediction performance on five other datasets of different species. They also validated the applicability of PLM-interact to identify the impact of mutations on interactions. While potentially interesting, I have the following concerns:

1. The PPI datasets were downloaded from the STRING V11 database. The authors should specify the type of these interactions, e.g. physical interactions, functional interactions and how different types might influence the model performance. In addition, is it possible that the datasets contain false positives and false negatives? If so, how to eliminate their influences?

Re: Thank you for your question. Overall, we have a large training set ~420k protein pairs. Errors in these data will not affect the quality of the model as long as they are not disproportionately abundant.

For positive pairs, only direct physical interactions from STRING V11 were selected. All other interaction types were excluded precisely for minimising the risk of false positive signals.

Following standard practice in the field, the negative pairs were randomly chosen from protein pairs that are not reported to be interacting. While it is possible we have included undiscovered interactions ie false negatives, it is unlike that there are a lot of them. Protein-protein interaction network is understood to be sparse.

As another mitigation strategy, we trained our model with a 1:10 positive to negative pairs ratio, such that a single potentially false negative would not be treated equally with a positive sample. This follows the training of similar models for other tasks, where there are only sparse true labels, and some unlabelled documents may be false negatives [Mikolov et al negative sampling: <https://arxiv.org/pdf/1310.4546>].

2. The design of the input sentence for PLM-interact is simple and direct. However, a major concern is that the number of proteins currently available is stable and thus a pretrained language model can be used to extract features in advance. For PLM-interact, given new pairs of proteins, the model needs to be retrained, which limits its practical usage.

Re: We apologise for the confusion. We want to clarify that PLM-interact does not need to be retrained to make predictions on new protein pairs. Our model can readily predict new pairs as long as the amino acid sequences of the proteins are provided. The cross-species benchmark (presented in Figure 2 and supplementary Figure 3) is closely related to the scenario envisioned by the reviewer. The protein pairs in those non-human species were not included in the training set. PLM-interact can still achieve state-of-the-art performance in this benchmark.

3. While the prediction performance seems to be astonishing, the baseline models selected are somewhat outdated and thus it is difficult to accurately evaluate the contribution of the PLM-interact.

Re: Thanks for your question. In this revision, we added TUnA, which is a PPI model published in September 2024. The revised benchmarking results are presented in updated Figure 2 and Supplementary Figure 3. We report that PLM-interact still outperforms TUnA across all species in the multi-species benchmark.

Both updated Figure 2 and Supplementary Figure 3 are pasted in the following:

a The training, validation and test PPI datasets

Figure 2. The benchmarking results of PLM-interact compared with state-of-the-art protein-protein interaction (PPI) prediction models: PLM-interact achieves the highest PPI prediction performance. **a.** The data size of training, validation and test protein pairs. **b.** The taxonomic tree of the training and test species is aligned with the precision-recall curve of each model on each test species. A bar plot of AUPR values illustrates the PPI prediction benchmark. The distribution of predicted interaction probabilities of positive and negative protein pairs for

each PPI model is shown in **Supplementary Figure 3**. All species icons in panel b are created in BioRender (Liu, D., 2025; <https://BioRender.com/1ezsj7q>).

Supplementary Figure 3. The distribution of prediction interaction probability of positive and negative protein pairs of PLM-interact, TUnA, TT3D, Topsy-Turvy and D-SCRIPT. PLM-

interact outperforms other models by identifying the most number of true positive pairs (predicted interaction probability > 0.5) and true negative pairs (predicted interaction probability < 0.5), demonstrating that PLM-interact achieves the best performance, except for negative pairs of *E. coli*.

4. The authors should conduct comparisons by using commonly used classifiers with the learned embeddings.

Re: We thank the reviewer for the suggestion. Most baselines included in our benchmark already follow the suggested classifier plus pretrained embedding approach. A well-known example is D-SCRIPT. PLM-interact differs from this approach by extending ESM-2's context length to accommodate a pair of pairs and fine-tune it with training protein pairs. We hope the reviewer will agree that being different from the suggested approach is the novelty of our method.

Reviewer #2 (Remarks to the Author)

The authors proposed a protein interaction prediction method, PLMsearch, that extends single protein-focused PLMs (ESM-2) to their interacting protein partner. Below are some questions I hope the authors can address:

1. The author's description of the method is not clear. For example, how exactly did the author fine-tune ESM-2? Specifically, does this process only involve modifying the parameters of the last layer of ESM-2, or were other aspects of the model also adjusted?

Re: We fine-tuned all layers of ESM-2 and a fully connected classification layer. In addition, the fine-tuning is done with the extended two-protein context. We added text to the PLM-interact section to explain that we update all layers of ESM-2. Please see details in PLM-interact section (see Line 68).

“To directly model PPIs, two extensions to ESM-2 are introduced: (1) longer permissible sequence lengths in paired masked-language training to accommodate amino acid residues from both proteins; (2) implementation of the “next sentence” prediction task to fine-tune all layers of ESM-2 where the model is trained with a binary label indicating whether the protein pair is interacting or not (see Methods for more details).”

2. What is the difference between the Masked Language Modeling (MLM) task used in PLM-Interact and ESM-2? Are there any experimental results demonstrating that performance is optimal when random masking is set at 15%?

Re: The masked language modelling in PLM-interact and ESM-2 is the same; both use random masking, and a 15% masking percentage, which is a common choice in training transformer encoders across application domains. However, the input to ESM-2 is a single protein, while PLM-interact follows the cross-encoder setup in that the input is 2 proteins. In this revision, we added an analysis of the model's performance under the different masking percentages. We trained additional models with varying masking ratios (0%, 7%, 15%, 22% and 30%,) on the human PPI training set and tested their AUPR performances in the cross-species benchmark. All models used the 650M ESM-2 architecture and a classification-to-mask loss ratio of 10:1.

Supplementary Figure 2. The AUPR values of the PLM-interact training with the different masking percentages on each test species. The x-axis represents the percentage of masking; the y-axis represents the AUPR value of each model on each test species. The gold star represents that the masking model has a significant improvement over the binary model (without masking), and p-value < 0.05 with McNemar’s Test.

We report the results in Supplementary Figure 2, which we also pasted here. Overall, the varying masking proportions does not introduce significant variations to the performance. To select the best option, we asked whether the masked version significantly outperforms the 0% no-masking binary baseline. Significant improvements based on McNemar’s test (a significance test often used for comparing classification outcomes) are marked by stars in Supplementary Figure 2. The result showed that the model with a 15% masking percentage is the only masking model that consistently outperforms the no-masking binary model. We hope these results reassure the reviewer that our masking option is sensible and the model’s performance is robust to different masking percentages.

3. The article mentions that conventional PLM-based PPI predictor architectures typically include a classification head, but PLM-Interact appears to lack one. Could the authors provide more detailed information regarding the classification-related aspects. In particular, please explain the design of the loss function in more detail.

Re: PLM-interact does have a classification head. The classification head takes the [CLS] token of the final layer of ESM-2 model as input features. The classification head is implemented as a fully connected network with a sigmoid activation. The setup is

equivalent to a logistic regression classifier. During training, we mask the amino acids of protein pairs and then obtain the masking loss. We used the combined masking and classification loss to optimize the model.

See Line 417, the loss function formular is:

$$l = \alpha l_{mlm} + \beta l_{ce},$$

We explain the classification head and loss function design in “Model Training” section.

4. For PPI prediction, many methods use human proteins as the training set and datasets from mouse, fly, worm, yeast, and E. coli as the testing set. However, is there sequence redundancy between these five datasets and the amino acid sequences of human proteins? Could an experiment be conducted to verify whether the model’s performance remains strong under different levels of redundancy between the training and testing sets?

Re: In this revision, we’ve included the experiment proposed by the reviewer. We evaluate PLM-interact’s performance under the different levels of protein sequence similarity between test protein pairs from five species and human training proteins. We found that, as expected, sequence similarity does play an important role in performance. At every level of similarity, PLM-interact outperforms the second-best approach, TUnA.

To evaluate sequence redundancy between human training and test five species, we use MMSeq2 to obtain sequence identities between training and test protein pairs on each test species, which are range from 0 to 100. The maximum sequence identity of any proteins from the test protein pair with training proteins is considered to be the sequence identity between the test protein pair and training proteins. Here, we assess the AUPR performance under the different levels of identity bins [0, 20, 40, 60, 80, 100]. To provide more context, we included TUnA to represent the behaviour of traditional two-branch architecture.

We summarise the results in the following figure, Supplementary Figure 7. For each panel, we report the AUPR (left y-axis) and the proportion of testing PPIs (right y-axis) vs the maximum protein sequence identifies.

Supplementary Figure 7. The AUPR performance of PLM-interact and TUnA under different

levels of protein identities between training and test datasets. The x-axis represents the identity of protein sequence between training and test proteins. For each panel, the left y-axis represents AUPR values, and the right y-axis represents the proportion of the test protein pairs under the corresponding protein identities (x-axis). The figure shows AUPR values of PLM-interact and TUnA on test protein pairs of mouse, fly, worm, yeast and *E. coli*, respectively. Their performance improves as protein identity increases, with PLM-interact consistently outperforming TUnA.

We add the following text to the main text (see Line 151):

“To probe the role of sequence identity in PLM-interact’s performance, we evaluate the model’s performance with the different protein identities between training and test datasets (see Method section for details). As expected, we observe marked sequence similarity between human and mouse, similarities reduce significantly between fly, worm, yeast and *E. coli* with human. Notably, both PLM-interact and the second-best performer in the cross-species benchmark benefit from sequence similarity. Their performance improves as protein identity increases, with PLM-interact consistently outperform TUnA (see Supplementary Figure 7).”

5. For virus-human PPI prediction, the authors re-used a virus-human PPIs dataset for model training, which seems to provide a model with different parameters. So, how does the model trained on more than 420,000 human proteins perform in virus-human PPI prediction? How is the generalization ability of the model?

Re: This is a great question. We observed that the model trained on human data performs less well on the virus-host data compared to the fine-tuned one. In general, viral proteins are very different from human proteins and are packed with disordered regions. In addition, viral diversities are not well represented in the pretraining data of ESM-2, due to the fact that ESM-2 was trained on representative sequences from sequence identity clusters such as Uniref90. These make the human PPI-trained model difficult to work for virus-human PPI prediction without fine-tuning.

6. Is the experimental purpose clearly defined, particularly in the case of the five positive PPI instances? The self-assessment score is mainly used to evaluate the accuracy of the predicted structure, rather than to evaluate whether the input sequence has interactions. It seems that the focus is more on explaining the inaccuracies in protein structure predictions from Chai-1 and AlphaFold3 rather than offering new insights into PPI prediction itself.

Re: We apologise for the confusion. Our goal is not to compare with structure prediction method but simply to visualise the SOTA predicted structures for the examples. We reported the self-assessment scores as part of standard practice. The fact that the scores are low underscores the complementary value of sequence-based PPI predictors. We have now moved self-assessment scores to table 2 in the supplementary material.

Figure 3. PPI example for each species that was predicted correctly by PLM-interact but not by TUnA and TT3D. Protein-protein structures are predicted by Chai-1²⁸ and visualised with ChimeraX³⁰. PPI models' prediction interaction probabilities range between 0 and 1. A predicted interaction probability >0.5, is considered to be a positive PPI, while <0.5 is a negative pair. Interacting proteins are shown from left (yellow) to right (green), respectively. **Mouse:** P97287 (Induced myeloid leukemia cell differentiation protein Mcl-1 homolog) and P63028 (Translationally-controlled tumor protein); **Fly:** Q9W0F0 (Dynein light chain roadblock) and Q7K035 (AT23443p); **Worm:** Q21955 (Mediator of RNA polymerase II transcription subunit 15) and Q9N4F2 (Mediator of RNA polymerase II transcription subunit 19); **Yeast:** P23644 (Mitochondrial import receptor subunit TOM40) and P53507 (Mitochondrial import receptor subunit TOM7); and **E. coli:** A0A454A7G5 (ABC transporter permease protein) and A0A454A7H5 (Possible ABC-transport protein, ATP-binding component). See Supplementary Figure 4 for the corresponding AlphaFold3²⁹ predicted structures. The ipTMs of both Chai-1 and AlphaFold3 for each structure are shown **Supplementary Table 2**.

Reviewer #3 (Remarks to the Author):

Overview

The authors introduce PLM-interact, an adaptation of the ESM-2 protein language model fine-tuned with protein-protein interaction (PPI) pairs to predict interactions given only the sequence of each protein. The model is supervised with two objectives, a masked language modeling objective over the paired sequence (separated by EOS tokens) and a classification accuracy (with a small classification head taking a leading CLS token as input), with the balance between these objectives chosen by a hyper-parameter search. They report strong performance compared to previous state-of-the-art methods on a multi-species PPI data set and on a virus-human host PPI benchmark set. They also compare to structure based methods AlphaFold3 and Chai-1, showing sensitivity to gain-of-function and loss-of-function mutations.

Sequence-based protein-protein interaction prediction is an important and well studied problem, especially the application of protein language models thereof. The question of whether language model pre-training can be adapted to include PPIs is well-motivated and the results are compelling. The results on gain- and loss-of-function mutations are especially interesting, since most previous literature has shown limited ability of protein language models to response to single mutations. The manuscript is overall well-written with a diverse set of experiments. The manuscript contains several valuable insights, such as the analogue to next-sentence prediction in natural language, and the fact that the performance gain of PLM-interact comes largely from the improvement in predicting positive PPIs, rather than improvement in the negative case. The authors develop a new PPI data set based on STRING v12 and release trained models based on this new, large-scale PPI data set. The code base appears clean and well-maintained, with clear training and inference scripts, and pre-trained models are made available through HuggingFace.

Re: We thank the reviewer for such a thorough summary of the contribution of the study.

The manuscript is currently lacking comparisons with recent state-of-the-art methods and benchmarks, as well as details about how train/test splits were determined that make it difficult to fully assess model performance. Relatedly, several of the biological examples seem arbitrarily picked and the manuscript would benefit from summary statistics, or at least a justification for how the case studies were chosen. Finally, the statistical test used for selection of objective balance hyper-parameters is either fully incorrect or improperly specified, these results likely contain data leakage from the test sets, and the results are not clearly communicated. We provide more details on these and other comments below.

Re: We thank the reviewer for pointing out the weaknesses of the current manuscript. In this revision, we have conducted extensive additional experimentation, including extra method and dataset for benchmarking, completely reworked mutation effect prediction section, and several technical assessments (sequence similarity, masking percentage, protein order swapping). We have improved our description of each dataset and its corresponding training, validation and test sets split. We have now clarified the role of

the examples of PPIs highlighted in the paper. We do not claim any superiority over structure prediction tools. In this revision, every application has a systematic performance assessment accompanied by examples of successful prediction. Finally, we have corrected misstatements about the hyper-parameter selection and statistical test. The detailed reply to each point raised by the reviewer can be found below. We hope our revision can address the concerns.

Major Comments

1. The authors should evaluate their method on the widely-used gold standard benchmarks from Bennett et al. 2024 ("Cracking the black box..."), the D-SCRIPT data set is a good measure of cross-species performance but is known to have some data leakage from homologous proteins

Re: Thank you for your suggestion. We have now included a benchmark on the Bennett human PPI dataset (Bennett et al. 2024). We trained PLM-interact ESM2-650M with a loss ratio of 1:10 for 5 epochs on this gold standard training set and used the validation set to choose the model for finally testing on the test set. The result is shown in Figure 4. PLM-interact and TUnA have very similar performance. They outperform other methods. PLM-interact performs better in recall, indicating relative strength in detecting positive interactions. Notably, the Bennett training dataset is small (1/3 of the cross-species training set from Sledzieski et al.). It puts PLM-interact, which requires many more parameters to be fine-tuned in a disadvantage.

Figure 4. The comparison of PPI models on the Bennett benchmarking dataset. a. The x-axis shows the evaluation metrics (AUPR, Precision, Recall, AUROC and F1-score) and y-axis represents the values of the metrics. **b.** The distribution of predicted interaction probabilities of PLM-interact and TUnA on the positive and negative protein pairs, respectively.

We added the following text to line 159 of the main text:

“To further evaluate our model’s performance in relation to sequence similarity, we train PLM-interact on a leakage-free human gold standard training dataset created by Bennett et al. and compare with the state-of-the-art PPI approaches. In this benchmarking dataset, there are no protein overlaps and minimal sequence similarities among training, validation and test datasets.

Due to computing limitations on the maximum sequence length of a pair, we only use 80% of training set for training. The result on the test set is shown in Figure 4, where PLM-interact exhibits the identical AUPR (0.69) and AUROC (0.7) with TUnA. Interestingly, when adopting a neutral 0.5 threshold on predicted interaction probabilities for final classification, PLM-interact outperforms TUnA and other baselines in F1-score and recall. The improvement in recall is 9% over TUnA, while the precision is comparable with TUnA, indicating PLM-interact performs better at predicting positive interactions.”

2. The authors should also consider comparing to RAPPID and TuNA, two state-of-the-art sequence-based PPI methods.

Re: Thank you for your suggestion. We have now included TUnA for both cross-species (Sledzieski et al. 2021) and the Bennett 2024 dataset.

We chose not to include RAPPID as the model adopts the traditional two-branch architecture, which recent implementations (post 2023) such as D-SCRIPT, Topsy-Turvy, TT3D and TUnA all follow. The key difference is RAPPID uses an LSTM for protein feature extraction, whereas later approaches adopt pretrained protein language models similar to our choice of protein encoders. Moreover, given that our key novelty is bringing inter-protein interaction signal to pretrained protein language models, we therefore assess that approaches using a pretrained protein language model are more appropriate baselines.

In the revised Figure 2, we added TUnA's AUPR results on mouse, fly, worm, yeast and *E. coli* obtained from the TUnA paper. Here we show the precision-recall curves of PPI models by recomputing the predicted interaction probabilities. TUnA reports three trained models, we load the checkpoint with the highest AUPR to plot this precision-recall curve for benchmarking.

a The training, validation and test PPI datasets

b

Figure 2. The benchmarking results of PLM-interact compared with state-of-the-art protein-protein interaction (PPI) prediction models: PLM-interact achieves the highest PPI prediction performance. **a.** The data size of training, validation and test protein pairs. **b.** The taxonomic tree of the training and test species is aligned with the precision-recall curve of each model on each test species. A bar plot of AUPR values illustrates the PPI prediction benchmark. The distribution of predicted interaction probabilities of positive and negative protein pairs for each PPI model is shown in **Supplementary Figure 3**. All species icons in panel b are created in BioRender (Liu, D., 2025; <https://BioRender.com/1ezsj7q>).

3. The structure-based experiments in Figure 3 are very interesting, but the authors should justify how these examples were chosen apart from others so as to avoid the appearance of cherry picking; or run AlphaFold3/Chai-1 on a wider test set to generate a summary of their performance.

Re: The goal of Figure 3 is to give qualitative examples (ie predicted structures) of PPIs from the cross-species benchmark in Figure 2. The examples are selected based PPI differential predictions from PPI predictors.

We do not claim that our PPI predictor outperforms structure prediction tools on PPI prediction. As pointed out by Reviewer 2, structure prediction tools don't answer the question of whether two molecules interact. Displaying the self-assessment (ipTM) scores from Chai-1 and AF3 could mislead readers. In the revision, we removed the ipTM scores from Figure 3 and reported them in Supplementary Table 2 instead. We hope the revised version makes the message from Figure 3 clearer.

The following is the revised Figure 3.

Figure 3. PPI example for each species that was predicted correctly by PLM-interact but not by TUnA and TT3D. Protein-protein structures are predicted by Chai-1²⁸ and visualised with ChimeraX³⁰. PPI models' prediction interaction probabilities range between 0 and 1. A predicted interaction probability >0.5, is considered to be a positive PPI, while <0.5 is a negative pair. Interacting proteins are shown from left (yellow) to right (green), respectively. **Mouse:** P97287 (Induced myeloid leukemia cell differentiation protein Mcl-1 homolog) and P63028 (Translationally-controlled tumor protein); **Fly:** Q9W0F0 (Dynein light chain roadblock) and Q7K035 (AT23443p); **Worm:** Q21955 (Mediator of RNA polymerase II transcription subunit 15) and Q9N4F2 (Mediator of RNA polymerase II transcription subunit 19); **Yeast:** P23644

(Mitochondrial import receptor subunit TOM40) and P53507 (Mitochondrial import receptor subunit TOM7); and *E. coli*: A0A454A7G5 (ABC transporter permease protein) and A0A454A7H5 (Possible ABC-transport protein, ATP-binding component). See Supplementary Figure 4 for the corresponding AlphaFold3²⁹ predicted structures. The ipTMs of both Chai-1 and AlphaFold3 for each structure are shown **Supplementary Table 2**.

4. Likewise, the choice of structure-based experiments in Figure 4 should be justified. Especially for mutation sensitivity, the authors should show how the aggregate performance of PLM-interact across the whole InAct data set compares to other sequence-based methods like Topsy-Turvy or PIPR (it would also be good to see AF3/Chai-1, but inference at this scale may not be possible).

Re: We thank the reviewer for the suggestion. We agree with the reviewer that the original version lacks a systematic assessment of the predicted mutation effect on PPIs. In this revision, we included a substantial study on predicting mutation effect PPIs using IntAct mutation data that are labelled as increasing (MI: 0382) or decreasing (MI: 0119) interaction rate or strength. In total, the two groups contain 6,979 annotated interactions with a decreasing vs increasing ratio of 4.5:1.

We examined models in both zero-shot and fine-tuning settings. In the zero-shot setting, we included TUnA, Topsy-Turvy and D-SCRIPT. In the fine-tuning setting, we proposed a novel fine-tuning method and compared two strategies, namely fine-tuning the entire PLM-interact model and fine-tuning the final classification layer. The latter is designed to mimic the conventional architecture (such as D-SCRIPT and others compared in our benchmarks), in which only a classifier head is trained.

We reported this systematic benchmark in Figure 5 and two successful cases in Figure 6. We have completely rewritten the mutation effect section.

The section “Fine-tuned PLM-interact can identify the impact of mutations on interactions” in the main text (from line 176) is the reworked mutation effect study.

In the Method section, we have included details of the dataset “Mutation effect dataset” (line 346-356), inference “Inference in the binary mutation effect task” (line 474-483), and fine-tuning “Fine-tuning in the binary mutation effect task” (line 484-497).

a Binary mutation effect classification task

b Inference and fine-tuning for mutation effect prediction

Figure 5. Predicting mutation effects on protein-protein interactions (PPIs). **a.** Diagrammatic overview of the binary mutation effect classification task. **b.** Inference and fine-tuning of PLM-interact to predict mutation effects that increase or decrease interaction rate/strength. The log formula in this panel is the log-predicted interaction probability ratio between the mutant and canonical pairs. **c.** Precision-Recall curves of two fine-tuned PLM-interact models and four zero-shot models (PLM-interact, TUnA, Topsy-Turvy and D-SCRIPT) on the test dataset. **d.** The ROC curves of the fine-tuned PLM-interact models and four zero-shot models on the test dataset.

Figure 6. Demonstration of PLM-interact detecting changes in human PPIs associated with mutations. **a** shows one mutation-increasing (ie mutation causes an increase in interaction) example, while **b** shows one mutation-decreasing (ie mutation causes an decrease in interaction) example. These PPI structures are predicted using Chai-1 and visualised with ChimeraX; here, the mutated amino acids are highlighted in purple. In each panel, the log-predicted interaction probability ratio between the mutant and canonical protein pairs is shown for fine-tuned PLM-interact, TUnA and Topsy-Turvy, respectively. The positive log ratio indicates a mutation-increasing PPI, while the negative log ratio indicates a mutation-decreasing PPI. The ipTMs of both Chai-1 and AlphaFold3 for each structure are shown in **Supplementary Table 3**. Interacting protein structures are shown from left (yellow) to right (green): **a**. Residue 600 Tyrosine (Y) of P33993 (DNA replication licensing factor MCM7) is mutated to Glutamic Acid (E), increasing its interaction with P33992 (DNA replication licensing factor MCM5). **b**. Residue 151 Asparagine (N) of Q16595 (Frataxin, mitochondrial) is mutated to Alanine (A), decreasing interaction with Q9H1K1 (Iron-sulfur cluster assembly enzyme ISCU). See **Supplementary Figure 10** for the corresponding AlphaFold3 predicted structures.

5. The mutation impact data set contains human proteins, and PLM-interact was trained

on human proteins-- the authors should report which proteins in the test set were also seen during training, and report the performance on mutational impact binned by similarity to any protein in the training set. Otherwise, there is a risk that the model is just memorizing true human PPIs and would not generalize to mutations further from this training set.

Re: We appreciate the concern about the risk of memorising. The reality is that most PPIs with mutation data are human. Therefore, many mutant proteins do have their wild type version in the training set. However, we found that overlapping proteins do not help in this case. Our zero-shot benchmark clearly shows that all models struggle. Only directly fine-tuning on the mutation effect data improves the performance.

Notably, in most cases, there is only a single amino acid difference between the canonical and mutant pairs, so similarity is very high. However, when a mutation causes a decrease in rate or strength (which is the most common label), the model is required to make an opposite prediction despite using almost identical sequences. Therefore, the model must find something beyond sequence similarity to solve the task. Looking forward, mutation effect prediction could be a valuable task to push models to look for non-similarity-based features. In this pursuit, our results provide useful state-of-the-art performance.

6. There is no information provided about how the HPIDB is split into training and test, nor how hyper-parameters were chosen without a validation set. If the splits from the original paper were used (since comparison metrics were taken from that paper), this should be specified.

Re: As the reviewer stated, the exact data split from Tsukiyama et al. 2021 was adopted for benchmarking. We included a description in the dedicated dataset section "Virus-human PPI dataset" (see Line 363-364).

The hyper-parameters were identical to previous training settings in this paper. Given that there is no validation set in Tsukiyama et al. 2021, we used the model trained after 10 epochs.

7. Since both protein orders (P1 EOS P2 EOS, P2 EOS P1 EOS) were provided at training time but not at inference time, how was the order chosen for validation and testing? Does the model make the same predictions if the order changes? It would be good to provide the model with both orders and average predictions if this value is not identical regardless of order.

Re: We thank the reviewer for this intriguing question. To thoroughly answer this question, we trained the model with two orders (P1-P2 and P2-P1). To check if the specific order plays a role in prediction, we tested the performance after we swapped the order in the inference setting in the cross-species test set and the human validation set. We found AUPR and AUPR-reverse are virtually identical. We also found that the distributions of the predicted interaction probabilities are almost identical after swapping the order. We have now included these in supplementary figures.

Supplementary Figure 5. The AUPR bar plot for PLM-interact shows the impacts of different orders of test protein pairs during inference. The x-axis represents five test species, and the y-axis represents AUPR values. The AUPR values are obtained by performing inference on the original and reverse order of the test protein pairs from each species.

Supplementary Figure 6. The distribution of predicted interaction probabilities between the original and reverse of each test protein pair from mouse, fly, worm, yeast and *E. coli*, respectively. The predicted interaction probabilities between the original and reverse order in each test protein pair exhibits almost identical distributions.

8. It is quite concerning that the values of alpha and beta were chosen based on test set performance with the other species, as this is the final value that is reported-- this is a large source of data leakage, as hyper-parameters should be chosen using a separate

validation set. Supp. Fig. 1 shows that the choice of these parameters does not drastically impact performance and the model performance is strong regardless, so this is not disqualifying, but these experiments should really be re-done on the validation set to accurately show the final test-set performance.

Re: The reviewer is correct in pointing out that we were not describing an optimisation procedure. We now report our comparison between alpha and beta value as a technical benchmark. We took the winner 1:10 and applied it in all downstream experiments. We have now changed the corresponding section title to “Technical benchmark for hyperparameter selection” (see Line 437).

Supplementary Figure 1. The benchmarking of different ratios of mask to classification loss on five species PPI prediction. **a.** The bar plots to show the ratio between mask loss and classification loss. **b** and **c** respectively represent the performance of our model with the different ratios between mask and classification loss on 650M and 35M of ESM-2 models. The x-axis shows AUPR values and y-axis is aligned with the taxonomy tree of the test hosts.

9. The McNemar's test is an uncommon statistical test that isn't very well motivated, and I believe it is improperly implemented. None of my institutions have access to the original manuscript, but Wikipedia, NE Hawass British Journal of Radiology 2014, and the statsmodels implementation are all in agreement with one another that the formula should be $\frac{(b - c)^2}{(b + c)}$, where b is the count of candidate pairs where model 1 has true positives AND model 2 has false negatives, and c is the count of candidate pairs where model 1 has false negatives AND model 2 has true positives. This is not

equivalent to what the authors provided, where $\frac{\text{model 1 true positives} + \text{model 1 true negatives}}{\text{model 2 true positives} + \text{model 2 true negatives}}$ is the `_ratio_` of (model 1 true positives + model 1 true negatives) to (model 2 true positives + model 2 true negatives). As a result, this test statistic is likely not Chi squared-distributed and the p-values are likely incorrect. I would recommend a more common statistical comparison, or the correct implementation of the McNemar's test to compare different models when re-doing model comparison using the validation sets.

Re: We thank the reviewer for spotting out the typographical error. We have now corrected the formula. To assure the reviewer we can confirm that the formula was correctly implemented in our codebase. The results are obtained by using McNemar's python package (<https://www.geeksforgeeks.org/how-to-perform-mcnemars-test-in-python/>). To the best of our knowledge, McNemar's test is well suited for comparing classification model performances.

10. Supp Fig 1 d, e is very confusing  what does "counts of better model" refer to here? If it is trying to show the relative ranking of each model (which I believe it is) by showing the number of other models it is more significant than, there are much better ways to show this data.

Re: We apologise about the confusion. We have removed Supplementary Figure 1d, e.

Minor Comments

- Line 52-54: "with a FFN being the dominant option, these classifiers often don't generalize well"  please provide a citation for this unsubstantiated claim.

Re: We have revised the statement to moderate the claim

"Unfortunately, with the use of 'frozen' embeddings and a feedforward neural network being the dominant option, these classifiers have limited parameters to deal with complex interaction patterns."

- Lines 124-126: "These PPIs are necessary... protein transportation"  please provide citations for these biological process associations for each protein.

Re: We added citations for each protein.

- Line 146: How were the thresholds of $ipTM < 0.6$ and $ipTM > 0.8$ chosen?

Re: It is based on the AlphaFold website recommendation. The threshold used here are from the guidance from the AlphaFold server (<https://alphafoldserver.com/faq#how-can-i-interpret-confidence-metrics-to-check-the-accuracy-of-structures>).

- Line 324: How were the learning rate, weight decay, etc. chosen?

Re: These hyperparameters are chosen from similar studies in the literature and our own trial and error experiments. Unfortunately, we don't have the compute resources to perform a systematic grid search for these hyperparameters.

The setting of these parameters is same as the cross-encoder training from the GitHub repository of sentence-transformer (https://github.com/UKPLab/sentence-transformers/tree/master/sentence_transformers/cross_encoder).

- Line 338: Please explain that details on selection of alpha and beta are described in a following section.

Re: We added the following text after the definition of alpha and beta. “alpha and beta are determined in the “Technical benchmark for hyperparameter selection”.

We revised text (line 424-425):

“Alpha and beta are weights for the MLM and classification losses, and they are determined in the following “Technical benchmark for hyperparameter selection” section.”

- Line 355: The difference between 0:1 (with mask) and 0:1 (without mask) is confusing-- if this is just whether or not the sequences are masked, why does with with mask task perform better than the pure classification task, which should have more information?

Re: Exactly, the difference is whether the sequences are masked. An important context to note is that these are settings during training only. The performance is assessed on held out inference setting where the trained models make predictions without any masking. Now, without masking certainly has more information as the model has the complete sequence. In the masking case, the model faces a tougher task to make the correct prediction with incomplete sequences. Masking in this case can be treated as a form of regulation to improve generalisation.

Typographical Errors

Line 28: add a comma after mutations.

Line 28: add a comma after virology.

We’ve corrected the error.

“Disruption of these protein-protein interactions (PPIs), e.g., mediated by mutations can underlie human disease. In virology PPIs are particularly.....”

Revised text:

“Disruption of these protein-protein interactions (PPIs), e.g., mediated by mutations, can underlie human disease. In virology, PPIs are particularly.....”

Line 56: specify that this context is lacking in model pre-training, not training

“To address the lack of inter-protein context in training, we propose....”

Revised text:

“To address the lack of inter-protein context in model pre-training, we propose....”

Eq. 7: model1_false, not fasle

We solved this problem.

Reviewer #3 (Remarks on **code availability**):

The code is well-written and well-documented, and the README provides clear instructions for installation of dependencies and the model itself, running training of the model, and performing model inference. The inference is somewhat clunky, as the user is required to define the model themselves; it would be good if the authors slightly cleaned up the inference procedure, but it is clean enough. I was able to download the model checkpoint and run inference. Due to availability of time/compute resources, I did not attempt to re-train the model myself or to replicate validation metrics with the trained model checkpoints.

Re: We thank the reviewer for checking our codebase. For this revision, we have prioritised revising the scientific content. We will improve our inference code as part of our continuous maintenance of the codebase.

REVIEWERS' COMMENTS

Reviewer #1 (Remarks to the Author):

The authors have addressed all my concerns.

Reviewer #2 (Remarks to the Author):

1、 The authors' description of the methodology is not sufficiently clear. For example, the fine-tuning of the ESM2 model and the use of the self-assessment score should be explained in more detail.

Details about ESM2 model training for PPI prediction are provided from in lines 413-437. Details about the fine-tuning of the ESM2 model for mutation effect prediction are provided in lines 498-511.

We added the following text at line 550 of the main text — a Self-Assessment Scores section— to explain the self-assessment scores pTM and ipTM:

Self-assessment scores

The ipTMs and pTM scores for the Chai-1 and AlphaFold3 predicted structures are reported in Supplementary Table 2 and Supplementary Table 3. ipTM scores below 0.6 indicate failed predictions, while scores above 0.8 indicate high confidence predictions. pTM scores above 0.5 suggest that the predicted structures are similar to the ground truth. For more details about these self-assessment scores, refer to the AlphaFold3 paper and official description at <https://alphafoldserver.com/faq#how-can-i-interpret-confidence-metrics-to-check-the-accuracy-of-structures>.

2、 For the virus-human PPI prediction, please provide experimental evidence.

In this virus–human PPI prediction task, positive PPIs are derived from experimentally validated interactions collected in the public dataset HPIDB 3.0 (<https://cales.arizona.edu/hpidb/>). However, there is no experimental evidence for negative PPIs in this task; they are randomly paired virus–human protein pairs that have not been reported as positive PPIs.

Reviewer #2 (Remarks on code availability):

The code is available.

Reviewer #3 (Remarks to the Author):

I thank the authors for their thoughtful responses to my comments, which they have suitably addressed in their revisions and in their response to reviews.

Reviewer #3 (Remarks on code availability):

The code is well-written and well-documented, and the README provides clear instructions for installation of dependencies and the model itself, running training of the model, and performing model inference. The inference is somewhat clunky, as the user is required to define the model themselves; it would be good if the authors slightly cleaned up the inference procedure, but it is clean enough. I was able to download the model checkpoint and run inference. Due to availability of time/compute resources, I did not attempt to re-train the model myself or to replicate validation metrics with the trained model checkpoints.

We updated the inference code to make it easier for using.